# Genome-wide association study of 23,500 individuals identifies 7 loci associated with brain ventricular volume

Dina Vojinovic et al.[#]

The volume of the lateral ventricles (LV) increases with age and their abnormal enlargement is a key feature of several neurological and psychiatric diseases. Although lateral ventricular volume is heritable, a comprehensive investigation of its genetic determinants is lacking. In this meta-analysis of genome-wide association studies of 23,533 healthy middle-aged to elderly individuals from 26 population-based cohorts, we identify 7 genetic loci associated with LV volume. These loci map to chromosomes 3q28, 7p22.3, 10p12.31, 11q23.1, 12q23.3, 16q24.2, and 22q13.1 and implicate pathways related to tau pathology, S1P signaling, and cytoskeleton organization. We also report a significant genetic overlap between the thalamus and LV volumes ($\rho_{\text{genetic}} = -0.59$, *p*-value $= 3.14 \times 10^{-6}$), suggesting that these brain structures may share a common biology. These genetic associations of LV volume provide insights into brain morphology.

The volume of lateral ventricles increases in normal aging[1–4]. The enlargement of lateral ventricles has also been suggested in various complex neurological disorders such as Alzheimer's disease, vascular dementia, and Parkinson's disease[5–8] as well as psychiatric disorders such as schizophrenia and bipolar disorder[9–11]. Furthermore, ventricular enlargement has been associated with poor cognitive functioning and cerebral small vessel disease pathology[12–14]. Even though it might be intuitive to interpret ventricular expansion primarily as an indicator of brain shrinkage after the onset of the disorder, recent studies have provided evidence against this notion[15,16]. The size of lateral ventricles is influenced by genetic factors with heritability estimated to be 54%, on average[16], but changing with age, from 32–35% in childhood to about 75% in late middle and older age[16]. Even though the size of surrounding gray matter structures is also heritable[17–19], ventricular volume is reported to be genetically independent of other brain regions surrounding the ventricles[20]. Similarly, ventricular enlargement in schizophrenia does not appear to be linked to volume reduction in the surrounding structures[15].

Elucidating the genetic contribution to inter-individual variation in lateral ventricular volume can thus provide important insights and better understanding of the complex genetic architecture of brain structures and related neurological and psychiatric disorders. Candidate gene studies have identified single-nucleotide polymorphisms (SNPs) mapping to Catechol-O-Methyltransferase (COMT) and Neuregulin 1 (NRG1) genes as associated with larger lateral ventricular volume in patients with the first episode of non-affective psychosis[21,22]. However, a comprehensive investigation of the genetic determinants of lateral ventricular volume is lacking.

Here, we perform a genome-wide association (GWA) meta-analysis of 23,533 middle-aged to elderly individuals from population-based cohorts participating in the Cohorts for Heart and Aging Research in Genomic Epidemiology (CHARGE) consortium in order to identify common genetic variants that influence lateral ventricle volume. We apply a commonly used two-stage GWA design followed by a joint analysis approach that combines information across the stages and provides greater power[23]. We identify 7 genetic loci associated with lateral ventricular volume and report genome-wide overlap with thalamus volume.

## Results

### Genome-wide association results
The overview of study design is illustrated in Supplementary Fig. 1. The GWA results from 12 studies were combined in stage 1 and subsequently evaluated in an independent sample from 14 studies in stage 2. Finally, the results of stage 1 and stage 2 analyses were combined in stage 3. Detailed information on study participants, image acquisition and genotyping is provided in Supplementary Note 1 and Supplementary Data 1–3.

The results of the stage 1 meta-analysis ($N = 11,396$) are illustrated in Supplementary Fig. 2. The quantile-quantile plot suggests that potential population stratification and/or cryptic relatedness are well controlled after genomic correction ($\lambda = 1.04$) (Supplementary Fig. 2, Supplementary Table 1). The stage 1 meta-analysis identified 146 significant variant associations, mapping to three chromosomal regions at 3q28, 7p22.3, and 16q24.2 (Table 1). All 146 stage 1 significant associations replicated in the stage 2 meta-analysis ($N = 12,137$) with the same direction of effect at Bonferroni adjusted significance ($p$-value $= 5 \times 10^{-3}$, Supplementary Data 4), except one SNP ($p$-value $= 7.6 \times 10^{-3}$). Subsequently, the results from all individual studies were combined in the stage 3 GWA meta-analysis ($N = 23,533$). The quantile–quantile plot

showed again adequate control of population stratification or relatedness (Supplementary Fig. 3). The combined stage 3 GWA meta-analysis identified 314 additional significant associations mapping to four additional chromosomal regions at 10p12.31, 11q23.1, 12q23.3, and 22q13.1 (Figs. 1, 2, Table 1). The effect size for the lead variant mapped to 10p12.31 locus was correlated with mean age of the cohort ($r = 0.50$, $p$-value $= 0.03$) (Supplementary Fig. 4). No correlation was found for the other lead variants (Supplementary Fig. 5–10).

Even though cohorts of European (EA) and African-American (AA) ancestry were included, all significant associations were mainly driven by EA samples (Supplementary Fig. 11–12). The direction of effect size across the EA cohorts for the seven lead variants was generally concordant and showed no evidence of any single cohort driving the associations (Supplementary Fig. 11). Despite the different methods of phenotyping across the cohorts, the cohorts with different phenotyping methods showed evidence of effect suggesting that there is limited heterogeneity in effects (Supplementary Fig. 12).

To investigate whether seven lead variants have an effect in early life, childhood, the analyses were carried out in a children's cohort of 1141 participants from Generation R study. The percentage of lead variants showing consistent direction of effect with stage 3 was 85.7% (6 out of 7, binomial $p$-value $= 0.05$) (Supplementary Data 4), and a variant mapped to the 12q23.3 region showed nominal association with lateral ventricular volume in the children's cohort (Zscore $= -2.56$, $p$-value $= 0.01$). Additionally, three out of seven lead variants (or their proxies; $r^2 > 0.7$) showed pleiotropic association ($p$-value $< 5 \times 10^{-8}$) with other traits according to the PhenoScanner database (Supplementary Data 5)[24].

To capture gender-based differences, sex-stratified GWA analysis was performed (Nmen $= 10,358$; Nwomen $= 12,872$). None of the 15,660,719 variants that were tested for heterogeneity between men and women reached genome-wide significance threshold (Supplementary Fig. 13). However, an indel located at 4q35.2 showed suggestive evidence of association in men (4:187559262:C_CAA, $p$-value $= 5.43 \times 10^{-8}$) but not in women ($p$-value $= 0.88$).

### Independent signals within loci
The conditional and joint (COJO) analysis using the Genome-wide Complex Trait Analysis (GCTA) identified no other additional variants, after conditioning on the lead variant at the locus 3q28, 7p22.3, 10p12.31, 11q23.1, 12q23.3, 16q24.2, or 22q13.1.

### Functional annotation
A large proportion of genome-wide significant variants were intergenic (335/460) (Supplementary Fig. 14). Variants with the highest probability of having a regulatory function based on RegulomeDB score (Category 1 RegulomeDB score) were located at 7p22.3 and at 22q13.1 (Supplementary Data 6). Of seven lead variants, four were intergenic, four were in an active chromatin state and three showed expression quantitative trait (eQTL) effects (Supplementary Data 6). The lead SNP at 22q13.1 (rs4820299) was associated with differential expression of the largest number of genes ($n = 6$). In brain tissue, the alternate allele of this SNP was associated with higher expression of TRIOBP suggesting that higher expression was associated with smaller lateral ventricles (Supplementary Fig. 15).

### Partitioned heritability
SNP-based heritability in the sample of European ancestry participants was estimated at 0.20 (SE = 0.02) using LD score regression, and this was higher in women 0.19 (SE = 0.04) than in men 0.15 (SE = 0.05). The seven lead variants

**Table 1 Genome-wide significant results from the meta-analyses of lateral ventricular volume**

| SNP | Chr | Annotation | Gene(s) | A1/A2 | Stage 1 | | Stage 2 | | Stage 3 combined | |
|---|---|---|---|---|---|---|---|---|---|---|
| | | | | | Zscore | P | Zscore | P | Zscore | P |
| rs34113929[a] | 3q28 | intergenic | *SNAR-I,OSTN* | A/G | −6.84 | 7.70E−12 | −5.05 | 4.44E−07 | −8.27 | 1.37E−16 |
| rs9937293[a] | 16q24.2 | intergenic | *FOXL1,C16orf95* | A/G | 5.65 | 1.63E−08 | 5.61 | 2.03E−08 | 7.84 | 4.45E−15 |
| 7:2760334:C_CT[a] | 7p22.3 | intergenic | *AMZ1,GNA12* | D/I | −5.88 | 4.21E−09 | −4.48 | 7.34E−06 | −7.21 | 5.61E−13 |
| rs12146713 | 12q23.3 | intronic | *NUAK1* | T/C | −5.01 | 5.57E−07 | −5.44 | 5.32E−08 | −7.28 | 3.25E−13 |
| rs4820299 | 22q13.1 | intronic | *TRIOBP* | T/C | −4.79 | 1.71E−06 | -4.49 | 7.04E−06 | −6.46 | 1.05E−10 |
| rs35587371 | 10p12.31 | intronic | *MLLT10* | A/T | −4.89 | 1.03E−06 | −3.32 | 9.12E−04 | −5.61 | 2.07E−08 |
| rs7936534 | 11q23.1 | intergenic | *ARHGAP20,C11orf53* | A/G | 4.25 | 2.12E−05 | 3.71 | 2.04E−04 | 5.54 | 2.96E−08 |

Variant that showed the lowest *p*-value in the fixed effect sample-size weighted *Z*-score meta-analysis for each locus is shown
SNP: single-nucleotide polymorphism, Chr: chromosome, A1/A2: effect allele/other allele, Freq: frequency of effect allele, Zscore: Z score from METAL, P: p-value
[a]Variants that surpassed genome-wide significance threshold in stage 1 meta-analysis; remaining SNPs listed in the table reached genome-wide significance threshold in combined, stage 3, meta-analysis

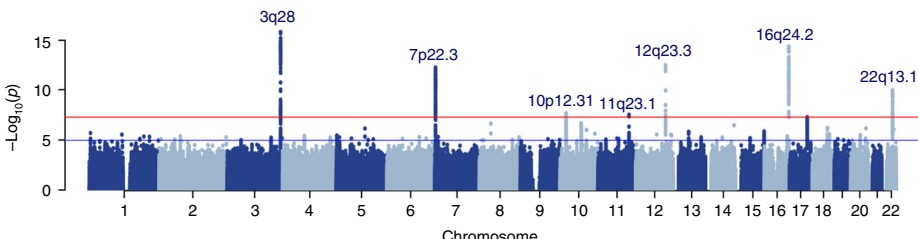

**Fig. 1** Manhattan plot for stage 3 genome-wide association meta-analysis. Each dot represents a variant. The plot shows –log10 *p*-values for all variants. Red line represents the genome-wide significance threshold ($p$-value $< 5 \times 10^{-8}$), whereas blue line denotes suggestive threshold ($p$-value $< 1 \times 10^{-5}$)

explained 1.5% of total variance in lateral ventricular volume. Partitioning of heritability based on functional annotation using LD score regression, revealed significant enrichment of SNPs within 500 bp of highly active enhancers, where 17% of SNPs accounted for 54% of the heritability ($p$-value $= 7.9 \times 10^{-6}$, Supplementary Table 2). Significant enrichment was also found for histone marks including H3K27ac (which indicates enhancer and promoter regions), H3K9ac (which highlights promoters), H3K4me3 (which indicates promoters/transcription starts), and H3K4me1 (which highlights enhancers) (Supplementary Table 2)[25,26].

**Functional enrichment analysis**. Functional enrichment analysis using regulatory regions from the ENCODE and Roadmap projects using the GWAS Analysis of Regulatory or Functional Information Enrichment with LD correction (GARFIELD) method revealed that SNPs associated with lateral ventricular volume at *p*-value threshold $<10^{-5}$ were more often located in genomic regions harboring histone marks (H3K9ac (associated with promoters) and H3K36me3 (associated with transcribed regions))[25] and DNaseI hypersensitivity sites (DHS) than a permuted background (Fig. 3, Supplementary Data 7).

**Integration of gene expression data**. Integration of functional data from the Genotype-Tissues Expression (GTEx) project using the MetaXcan method revealed two significant associations between genetically predicted expression in brain tissue and lateral ventricular volume (Supplementary Fig. 16). Expression levels of *TRIOBP* at the locus 22q13.1 ($p$-value $= 3.2 \times 10^{-6}$) and *MRPS16* at the locus 10q22.2 ($p$-value $= 1.8 \times 10^{-6}$) were associated with lateral ventricular volume.

**Gene annotation and pathway analysis**. The results of gene-based and pathway analyses are illustrated in Supplementary Table 3 and Supplementary Data 8. The pathway analysis

identified "regulation of cytoskeleton organization" (GO:0051493) gene-set to be significantly enriched ($p$-value $= 6 \times 10^{-6}$). Genes of the "regulation of cytoskeleton organization" pathway have previously been implicated in various neurological or cardiovascular diseases (Supplementary Data 9). Furthermore, pathways that pointed towards sphingosine 1 phosphate (S1P) signaling showed suggestive enrichment (Supplementary Data 8).

**Genetic correlation**. Additionally, we examined the genetic overlap between lateral ventricular volume and other traits (Table 2). We found that genetically determined components of thalamus and lateral ventricular volumes appear to be negatively correlated ($\rho_{genetic} = -0.59$, $p$-value $= 3.14 \times 10^{-6}$). This finding was also confirmed at the phenotype level (Supplementary Table 4). Weaker genetic overlap was observed with infant head circumference ($\rho_{genetic} = 0.28$, $p$-value $= 8.7 \times 10^{-3}$), intracranial volume ($\rho_{genetic} = 0.35$, $p$-value $= 9 \times 10^{-3}$), height ($\rho_{genetic} = -0.14$, $p$-value $= 5.7 \times 10^{-3}$), and mean pallidum ($\rho_{genetic} = -0.29$, $p$-value $= 2.5 \times 10^{-2}$), whereas no significant genetic overlap was found with neurological diseases, psychiatric diseases, or personality traits.

**Genetic risk score**. We next examined the association of genetic risk scores (GRS) for Alzheimer's disease, Parkinson's disease, schizophrenia, bipolar disorder, cerebral small vessel disease, and tau-related pathology, including tau and phosphorylated tau levels in cerebrospinal fluid, amyotrophic lateral sclerosis (ALS), and progressive supranuclear palsy (PSP), using the lead SNPs from the largest published GWA study and lateral ventricular volume (Supplementary Data 10). We found a suggestive association of GRS for tau levels in cerebrospinal fluid ($p$-value $= 9.59 \times 10^{-3}$) and lateral ventricular volume (Supplementary Table 5). The association was driven by one SNP (Supplementary

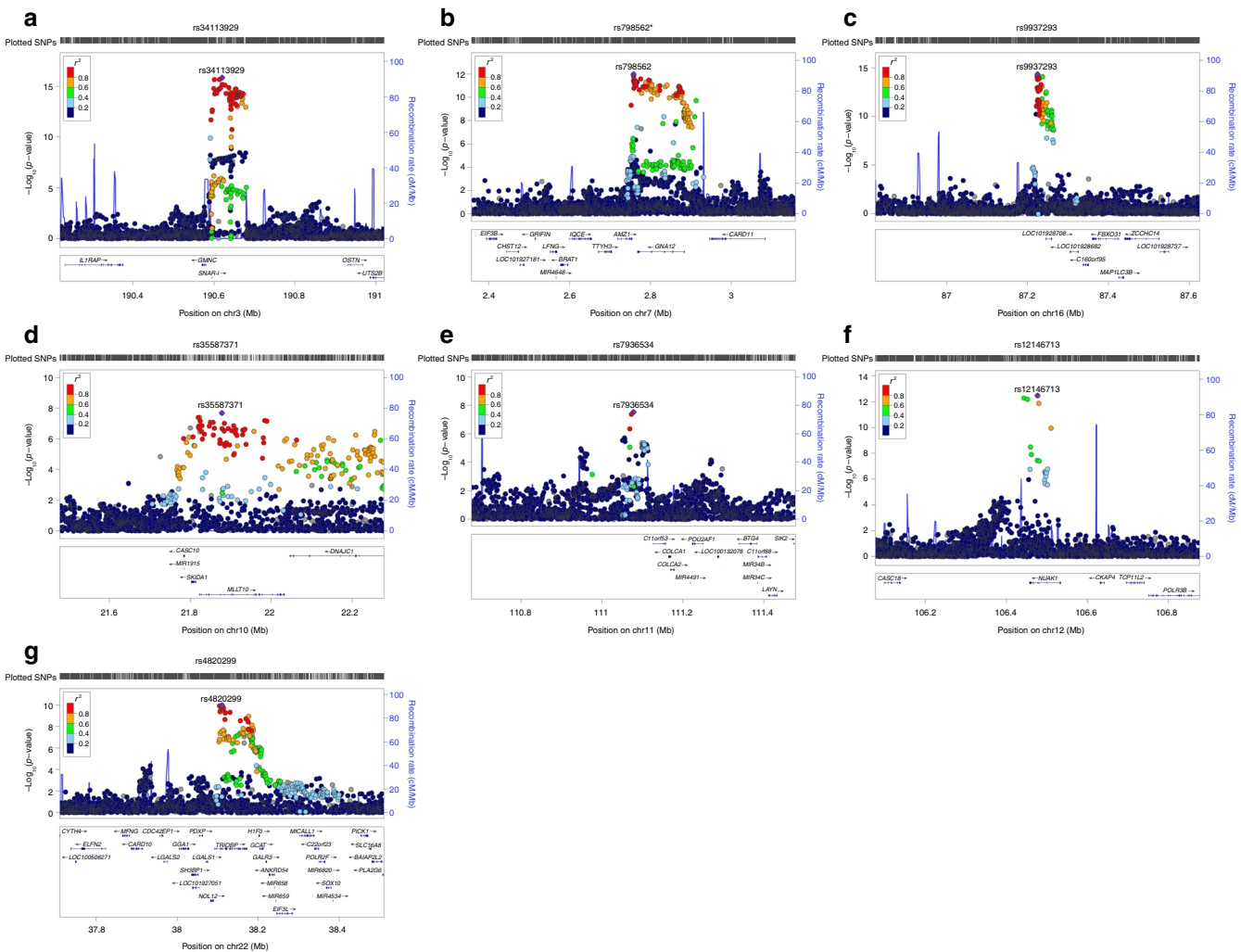

**Fig. 2** Regional association and recombination plots in combined stage 3 GWA meta-analysis. The left axis represents –log10 $p$-values for association with total later ventricular volume. The right axis represents the recombination rate, and the $x$-axis represents chromosomal position (hg19 genomic position). The most significant SNPs of the regions are denoted with a purple diamond. Surrounding SNPs are colored according to their pairwise correlation ($r^2$) with the top-associated SNP of the region. The gene annotations are below the figure

Table 6). No association was observed with other examined phenotypes (Supplementary Table 5).

## Discussion

We have performed the first genome-wide association study of lateral ventricular volume including up to 23,533 individuals. We identified statistically significant association between lateral ventricular volume and variants at 7 loci. Additionally, we found that genetically determined components of thalamus and lateral ventricular volume are correlated.

The strongest association was observed at the intergenic 3q28 locus between non-coding RNA *SNAR-I* and *OSTN*. This region has previously been associated with cerebrospinal fluid tau/ptau levels and Alzheimer's disease risk, tangle pathology and cognitive decline[27]. Similarly, the genome-wide significant locus at 12q23.3 encompasses *NUAK1*, which has also been associated with tau pathology. Nuak1 modulates tau levels in human cells and animal models and associates with tau accumulation in different tauopathies[28]. *NUAK1* is most prominently expressed in the brain where it has a role in mediating axon growth and branching in cortical neurons[29]. The lead SNP of the 12q23.3 locus mapped to an intron of *NUAK1*. This SNP is among the top 1% of most deleterious variants in the human genome based on its Combined Annotation Dependent Depletion (CADD) score of 21.5 and is located in an enhancer region (Supplementary Data 6). Interestingly, this variant also showed an effect in early life.

In our data, the significant variants of 7p22.3 region had the highest probability of being regulatory based on the RegulomeDB score (1b). The lead variant at 7p22.3 was in an active chromatin state and was associated with differential expression of *GNA12* (Supplementary Data 6). The *GNA12* gene is involved in various transmembrane signaling systems[30–33]. Interestingly, this gene was part of S1P signaling pathways identified to be enriched among genes associated with lateral ventricular volume. S1P, a bioactive sphingolipid metabolite, regulates nervous system development[34] such as neuronal survival, neurite outgrowth, and axon guidance[35,36], and plays a role in neurotransmitter release[37]. It also plays a role in regulating the development of germinal matrix (GM) vasculature[38]. Disruption of S1P regulation results in defective angiogenesis in GM, hemorrhage, and enlarged ventricles[38].

The other identified locus, 16q24.2, has previously been connected with small vessel disease and white-matter lesions formation[39]. Further, the alternate allele of the lead SNP at

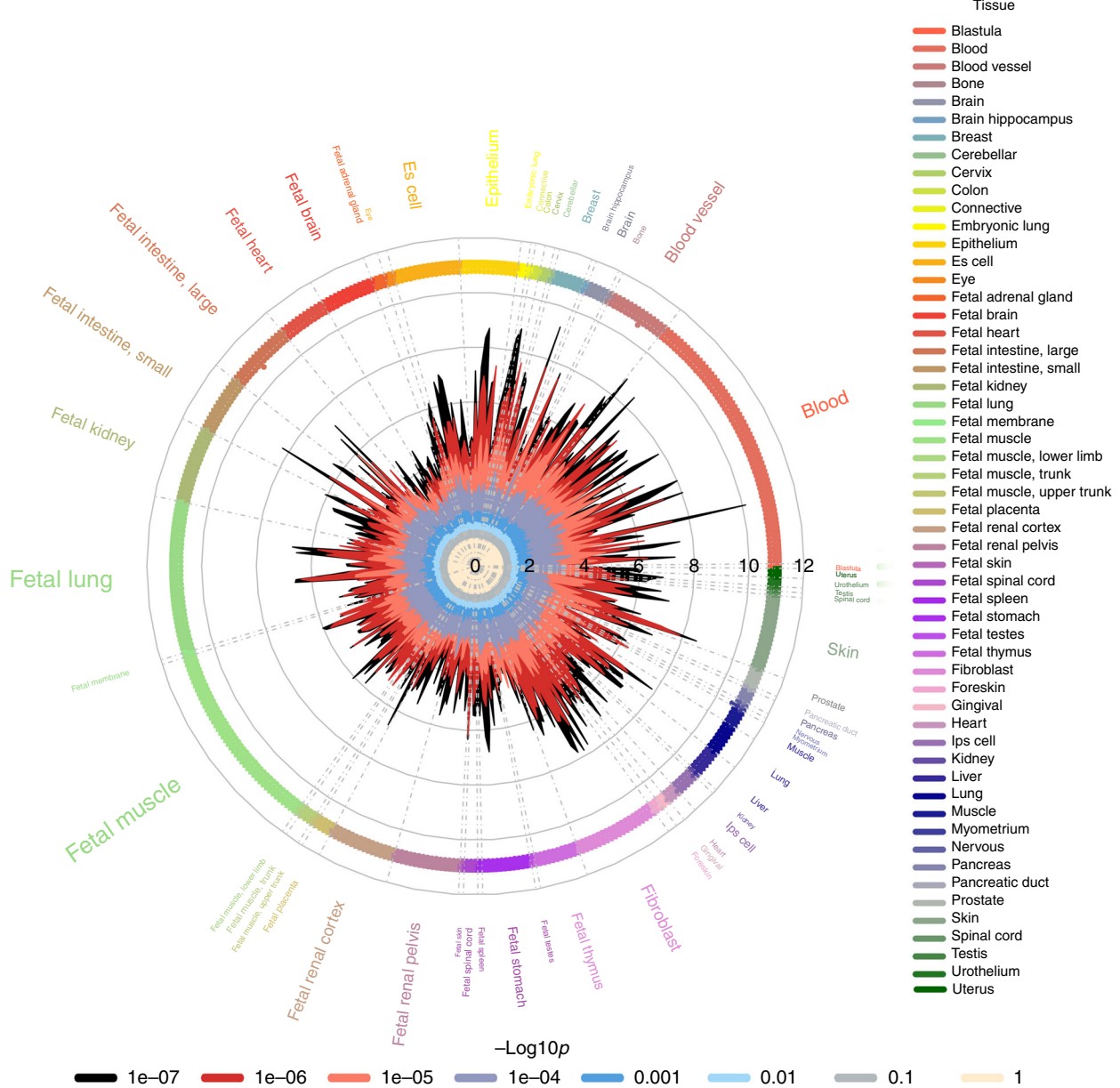

**Fig. 3** Functional enrichment analysis of lateral ventricular volume loci within DNaseI hypersensitivity spots. The radial lines show fold enrichment (FE) at eight GWA *p*-value thresholds. The results are shown for each of 424 cell types which are sorted by tissue, represented along the outer circle of the plot. The font size is proportional to the number of cell types from the tissue. FE values are plotted with different colors with respect to different GWA thresholds. Significant enrichment for a given cell type is denoted along the outer circle of the plot from a GWA *p*-value threshold $<10^{-5}$ (outermost) to GWA *p*-value threshold $<10^{-8}$ (innermost). The results show ubiquitous enrichment

22q13.1 in *TRIOBP* is associated with higher expression of the same gene in basal ganglia and brain cortex, and the same allele is associated with smaller lateral ventricular volume. Interestingly, predicted expression of this gene in cerebral cortex was significantly associated with lateral ventricular volume, suggesting a causal functional role of the gene. The same analysis revealed significant association of the expression of *MRPS16* in frontal cortex with lateral ventricular volume. This gene was previously related to agenesis/hypoplasia of corpus callosum and enlarged ventricles[40].

Finally, the lead intergenic SNP at 11q23.1 maps between *C11orf53* and *ARHGAP20*, whereas the 10p12.31 region encompasses *MLLT10* which has been linked to various leukemias, ovarian cancer, and meningioma[41,42]. The effect size of this

variant on lateral ventricular volume was correlated with mean cohort age, with the effect being near zero at younger age and larger at older ages.

The gene-enrichment analysis highlighted "regulation of cytoskeleton organization" (GO:0051493) pathway. Genes that are part of this pathway have previously been implicated in various neurological diseases such as Parkinson's disease (*PARK2*), frontotemporal dementia (*MAPT*), neurofibromatosis 2 (*NF2*), tuberous sclerosis (*TSC1*) (Supplementary Data 9). The cytoskeleton is essentially involved in all cellular processes, and therefore crucial for processes in the brain such as cell proliferation, differentiation, migration, and signaling. Dysfunction of cytoskeleton has been associated with neurodevelopmental, psychiatric and neurodegenerative diseases[43–45].

**Table 2 The results of genetic correlation between the lateral ventricular volume and anthropometric traits, brain volumes, neurological and psychiatric diseases and personality traits**

| Category | Phenotype | PMID | N | rg | SE | P |
|---|---|---|---|---|---|---|
| Anthropometric | | | | | | |
| | Height | 20881960 | 133,859 | −0.135 | 0.049 | 5.70E−03 |
| | Infant head circumference | 22504419 | 10,768 | 0.284 | 0.108 | 8.70E−03 |
| | Child birth length | 25281659 | 28,459 | −0.133 | 0.089 | 1.34E−01 |
| | Child birth weight | 23202124 | 26,836 | −0.118 | 0.102 | 2.47E−01 |
| Brain volume | | | | | | |
| | Mean thalamus | 25607358 | 13,193 | −0.591 | 0.127 | 3.14E−06 |
| | Mean pallidum | 25607358 | 13,142 | −0.29 | 0.129 | 2.47E−02 |
| | ICV | 25607358 | 11,373 | 0.347 | 0.133 | 9.00E−03 |
| | Mean accumbens | 25607358 | 13,112 | −0.29 | 0.158 | 6.64E−02 |
| | Mean putamen | 25607358 | 13,145 | −0.15 | 0.089 | 9.13E−02 |
| | Mean hippocampus | 25607358 | 13,163 | −0.204 | 0.132 | 1.20E−01 |
| | Mean caudate | 25607358 | 13,171 | 0.012 | 0.105 | 9.06E−01 |
| Neurological diseases | | | | | | |
| | Alzheimer's disease | 24162737 | 54,162 | 0.181 | 0.11 | 9.87E−02 |
| | Parkinson's disease | 19915575 | 5691 | −0.096 | 0.084 | 2.55E−01 |
| | Amyotrophic lateral sclerosis | 27455348 | 36,052 | −0.032 | 0.128 | 8.04E−01 |
| | White matter hyperintensities | 25663218 | 17,940 | 0.100 | 0.094 | 2.87E−01 |
| Personality traits | | | | | | |
| | Neo-conscientiousness | 21173776 | 17,375 | −0.359 | 0.158 | 2.27E−02 |
| | Neo-openness to experience | 21173776 | 17,375 | 0.088 | 0.118 | 4.56E−01 |
| | Neuroticism | 27089181 | 170,911 | −0.03 | 0.065 | 6.45E−01 |
| Psychiatric traits | | | | | | |
| | ADHD | 20732625 | 5422 | −0.276 | 0.152 | 6.90E−02 |
| | PGC cross-disorder analysis | 23453885 | 61,220 | −0.121 | 0.071 | 8.65E−02 |
| | Major depressive disorder | 22472876 | 18,759 | −0.165 | 0.102 | 1.05E−01 |
| | Schizophrenia | 25056061 | 77,096 | −0.067 | 0.044 | 1.30E−01 |
| | Subjective well-being | 27089181 | 298,420 | 0.087 | 0.075 | 2.50E−01 |
| | ADHD (no GC) | 27663945 | 17,666 | −0.151 | 0.149 | 3.11E−01 |
| | Depressive symptoms | 27089181 | 161,460 | −0.038 | 0.071 | 5.93E−01 |
| | Autism spectrum disorder | 0 | 10,263 | 0.041 | 0.092 | 6.53E−01 |
| | Anorexia nervosa | 24514567 | 17,767 | 0.011 | 0.056 | 8.43E−01 |
| | Bipolar disorder | 21926972 | 16,731 | 0.009 | 0.078 | 9.12E−01 |

rg: genetic correlation, SE: standard error, P: p-value

Previous studies showed significant sex-specific differences in lateral ventricular volume[46,47]. In our study we did not observe sex-specific differences; as for the lead seven variants, both males and females were contributing to the association signal. However, we observed only one suggestive association at 4q35.2 that showed association in men only. The lead variant (indel) is mapped to *FAT1* which encodes atypical cadherins. Mutation in this gene causes a defect in cranial neural tube closure in a mouse model and an increase in radial precursor proliferation in the cortex[48]. However, the SNP-based heritability estimates were slightly higher in females. This may be explained by the differences in sample size in male and female-specific analyses implying that there is lower precision.

We estimated that 20% of genetic variance in lateral ventricular volume could be explained by common genetic variants, suggesting that common variants represent a substantial fraction of overall genetic component of variance. Moreover, the most statistically significant effect occurred in the regions of highly active enhancers and histone marks, suggesting their involvement in gene expression. Using the LD score regression method, we found a significant negative genetic correlation between lateral ventricular volume and thalamus volume. However, these may not be independent events, but inverse reflections of the same biology. Even though not strictly significant, we also observed trends for genetic correlations with

other brain volumetric measures. Furthermore, no genome-wide overlap was found between lateral ventricular volume and various neurological or psychiatric diseases. Given that enlargement of lateral ventricles has been suggested in Alzheimer's disease, we examined the association of *APOE* alleles and found no association between the *APOE* ε4 (*p*-value = 0.86) or *APOE* ε2 (*p*-value = 0.81) and lateral ventricular volume in our study population.

As we identified loci underlying lateral ventricular volume at the genome-wide level, but also genes and common pathways, our results provide various insights into the genetic contribution to lateral ventricular volume variability and a better understanding of the complex genetic architecture of brain structures. The genes with variants that we found to be associated with lateral ventricular volume are relevant to neurological aging given the characteristics of the study population which is relatively free from the disease as participants with stroke, traumatic brain injury and dementia at the time of magnetic resonance imaging (MRI) were excluded. This is in line with the previously published work of Pfefferbaum et al. who showed that the stability of lateral ventricles is genetically determined, whereas other factors such as normal aging or trauma and disease play a role in its change[1,16].

However, while studying genetic overlap of lateral ventricular volume and various neurological or psychiatric disorders at

multiple levels (LD score regression/polygenic, GRS/oligogenic, GWA hits/monogenic), we found evidence that some single genetic variants have pleiotropic effect on lateral ventricular volume and biochemical markers for a neurological disease (AD) or meningioma (Supplementary Data 5), while no evidence was found for genetic overlap with other neurological or psychiatric disorders (Table 2, Supplementary Table 5). The pattern of association between lateral ventricular volume and psychiatric disorder, i.e., schizophrenia on multiple scales is similar to the findings of Franke et al. who evaluated association of various subcortical brain volumes and schizophrenia and reported no evidence of genetic overlap[49]. Even though our study does not provide a definite statement regarding the relationship between lateral ventricular volume and neurological or psychiatric disorders, it lays the foundation for future studies which should disentangle whether lateral ventricular volume is genetically related or unrelated to various neurological and psychiatric disorders (e.g., result from reverse causation). Novel insights may be revealed by improving the power of the studies, studying homogeneous samples with harmonized phenotype assessment methods along with evaluation of common and rare variants.

The strengths of our study are the large sample, population-based design and the use of quantitative MRI. Our study also has several limitations. Despite the effort to harmonize phenotype assessment, the methods used to quantify lateral ventricular volume differ across cohorts. Because of this phenotypic heterogeneity, association results of participating cohorts were combined using a sample-size weighted meta-analysis, thus limiting discussion on effect sizes. Secondly, phenotypic heterogeneity may have caused the loss of statistical power. However, despite heterogeneity in the phenotype assessment, the association signals were coming from several studies irrespective of the method of phenotype assessment, which suggests robustness of our findings. Furthermore, although we made an effort to include cohorts of EA and AA ancestry, the study comprised predominately of individuals of European origin (22,045 individuals of EA and 1488 of AA ancestry). Given the disparity in sample size, it is difficult to distinguish whether any inconsistency in results between the two groups stems from true genetic differences or from differential power to detect genetic effects. Indeed, this is also exemplified by the plots of the Z-scores (Supplementary Fig. 11) showing that direction of effect size in AA cohorts is often inconsistent with the direction of effect size in EA cohorts. However, the same inconsistency can be observed with European cohorts of equally small sample size. This inconsistency may be due to small sample size rather than ethnic background but we cannot rule out that racial-ethnic specific effects may exit. This limitation underscores the need for expanding research studies in non-European populations. Finally, as some loci only reached the genome-wide significance in the combined meta-analysis, they should be considered as highly probable findings and would still require independent replication.

To conclude, we identified genetic associations of lateral ventricular volume with variants mapping to 7 loci and implicating several pathways, including pathway related to tau pathology, cytoskeleton organization, and S1P signaling. These data provide new insights into understanding brain morphology.

## Methods

**Study design**. The overview of study design is illustrated in Supplementary Fig. 1. We performed a GWA meta-analysis of 11,396 participants of mainly European ancestry from 12 studies (stage 1) that contributed summary statistic data before a certain deadline. The deadline was set prior to data inspection and was not influenced by the results of the GWA meta-analysis. Variants that surpassed the genome-wide significance threshold (p-value $< 5 \times 10^{-8}$) were subsequently evaluated in an independent sample of 12,137 participants of mainly European ancestry from 14 studies (stage 2). Finally, we performed a meta-analysis of all stage 1 and stage 2 studies (stage 3).

**Study population**. All participating studies are part of the Cohorts for Heart and Aging Research in Genomic Epidemiology (CHARGE) consortium[50]. A detailed description of participating studies can be found in Supplementary Note 1. General characteristics of study participants are provided in Supplementary Data 1. Written informed consent was obtained from all participants. Each study was approved by local ethical committees or the institutional review boards (see Supplementary Note 1 for details).

**Imaging**. Each study performed MRI and estimated the volume of the lateral ventricles and intracranial volume (ICV). The field strength of scanners ranged from 0.35 to 3 T. Information on scanner manufacturers and measurement methods is provided in Supplementary Data 2. While most of the studies quantified lateral ventricular volume using validated automated segmentation methods, some studies used validated visual grading scales. The visual and volumetric scales were compared previously and showed high agreement for lateral ventricular volume[2]. The assessment of consistency of lateral ventricular volume on volumetric scale across time and different versions of software (freesurfer v4.5, v5.1, and v6.0), revealed high intraclass correlation (ICC > 0.98) in a subset of participants from the Rotterdam Study. Participants with dementia at the time of MRI, traumatic brain injury, prior or current stroke or intracranial tumors were excluded.

**Genotyping and imputation**. Information on genotyping platforms, quality control procedures and imputations methods for each participating study are provided in Supplementary Data 3. All studies used commercially available genotyping arrays, including Illumina or Affymetrix arrays. Similar quality control procedures were applied for each study (Supplementary Data 3). Using the validated software (Minimac[51], IMPUTE[52], BEAGLE[53]), each study performed genotype imputations using mostly the 1000 Genome phase 1 v3 reference panel.

**Genome-wide association (GWA) analysis**. Each participating study performed the GWA analysis of total lateral ventricular volume under an additive model using variant allele dosage as predictors and natural logarithm of the total lateral ventricular volume as the dependent variable. Transformation of the lateral ventricular volume was applied to obtain approximately normal distribution (Supplementary Fig. 17). The association analyses were adjusted for age, sex, total intracranial volume, age[2] if significant, population stratification, familial relationship (family-based studies) or study site (multi-site studies). Population stratification was controlled for by including principal components derived from genome-wide genotype data. Study-specific details on covariates and software used are provided in Supplementary Data 3. Quality control (QC) was conducted for all participating studies using a standardized protocol provided by Winkler et al.[54]. Variants with low imputation quality $r^2 < 0.3$ or minor allele count (MAC) $\leq 6$ were filtered out. The association results of participating studies were combined using a fixed-effect sample-size weighted Z-score meta-analysis in METAL because of the difference in measurement methods of lateral ventricular volume[55]. Genomic control was applied to account for small amounts of population stratification or unaccounted relatedness. After the meta-analysis, variants with information in less than half the total sample size were excluded. Meta-analyses were performed separately for each of the stages. In the stage 1 meta-analysis, a p-value $< 5 \times 10^{-8}$ was considered significant. Variants that surpassed the threshold were evaluated in the stage 2 meta-analysis. In order to model linkage disequilibrium (LD) between those variants, we first calculated the number of independent tests using the eigenvalues of a correlation matrix using the Matrix Spectral Decomposition (matSpDlite) software[56]. Subsequently, a Bonferroni correction was applied for the effective number of independent tests (0.05/10 independent SNPs = $5 \times 10^{-3}$). Additionally, all analyses were stratified by sex. Following the same QC steps as for overall analyses, the sex-stratified association results of participating studies were combined using a fixed-effect sample-size weighted Z-score meta-analysis in METAL while applying genomic control[55]. The variants were assessed only if test statistics (Z-score) were heterogeneous between males and females (p-value < 0.1) and if the association in a sex-combined analysis did not reach genome-wide significance threshold[57].

**Conditional analysis**. In order to identify variants that were independently associated with lateral ventricular volume, we performed conditional and joint (COJO) GWA analysis using Genome-wide Complex Trait Analysis (GCTA), version 1.26.0[58]. LD pattern was calculated based on 1000 Genome phase 1v3 imputed data of 6291 individuals from the Rotterdam Study I.

**Functional annotation**. To annotate genome-wide significant variants with regulatory information, we used HaploReg v4.1[59], RegulomeDB v1.1[60], and Combined Annotation Dependent Depletion (CADD) tools[61]. To determine whether they have an effect on gene expression, we used GTEx data[62]. For the lead variants, we explored 5 chromatin marks assayed in 127 epigenomes (H3K4me3, H3K4me1,

H3K36me3, H3K27me3, H3K9me3) of RoadMap data[63]. To search for pleiotropic associations between our lead variants and their proxies ($r^2 > 0.7$) with other traits, we used the PhenoScanner database designed to facilitate the cross-referencing of genetic variants with many phenotypes[9]. The association results with genome-wide significance at $5 \times 10^{-8}$ were extracted.

**Variance explained**. The proportion of variance in lateral ventricular volume explained by each lead variant was calculated using Pearson's phi coefficient squared as explained in Draisma et al.[64]. The total proportion of variance in lateral ventricular volume was calculated by adding up the proportions of variance in lateral ventricular volume explained by each lead association signal.

**Partitioned heritability**. SNP-based heritability and partitioned heritability analyses were performed using LD score regression following the previously described method[65]. Partitioned heritability analysis determines enrichment of heritability in SNPs partitioned into 24 functional classes as reported in Finucane et al.[65]. To avoid bias, an additional 500 bp window was included around the variants included in the functional classes. Only the HapMap3 variants were included as these seem to be well-imputed across cohorts.

**Functional enrichment analysis**. We performed functional enrichment analysis using regulatory regions from the ENCODE and Roadmap projects using GWAS Analysis of Regulatory or Functional Information Enrichment with LD correction (GARFIELD) method[66]. The method provides fold enrichment (FE) statistics at various GWA $p$-value thresholds after taking into account LD, minor allele frequency, and local gene density[66]. The FE statistics were calculated at eight GWA $p$-value thresholds (0.1 to $1 \times 10^{-8}$). The associations were tested for various regulatory elements including DNase-I hypersensitivity sites, histone modifications, chromatin states and transcription factor binding sites in over 1000 cell and tissue-specific annotations[66]. The significance threshold calculated based on the number of annotations used was set at $4.97 \times 10^{-5}$.

**Integration of gene expression**. To integrate functional data in the context of our meta-analysis results, we used the MetaXcan method, which evaluated the association between lateral ventricular volume and brain-specific gene-expression levels predicted by genetic variants using the data from GTEx project[62,67]. This method is an extension of PrediXcan method modified to use summary statistic data from meta-analysis[67]. Based on a total number of genes tested, the Bonferroni-corrected significance threshold was set to $0.05/12,379 = 4 \times 10^{-6}$.

**Gene annotation and pathway-based analysis**. The gene-based test statistics were computed using VEGAS2 software which tests for enrichment of multiple single variants within the genes while accounting for LD structure[68]. LD structure was computed based on the 1000 Genomes phase 3 population. Variants within 10 kb of the 5′ and 3′ untranslated regions were included in this analysis in order to maintain regulatory variants[68]. Subsequently, the gene-based scores were used to perform gene-set enrichment analysis using VEGAS2pathway[69]. VEGAS2Pathway approach accounts for LD between variants within a gene, and between neighboring genes, gene size, and pathway size[69]. It uses computationally predicted Gene Ontology pathways and curated gene-sets from the MSigDB, PANTHER, and pathway commons databases[69]. The pathway-based significance threshold was set to the $p$-value $= 1 \times 10^{-5}$ while taking into account the multiple testing of correlated pathways (0.05/5000 independent tests)[69].

**Genetic correlation**. We used the LD score regression method to estimate genetic correlations between lateral ventricular volume and various traits including anthropometric traits, brain volumes, neurological and psychiatric diseases and personality traits. The analyses were performed using a centralized database of summary-level GWA study results and a web interface for LD score regression, the LD-hub[70]. Summary-level GWA study results for white matter hyperintensities were obtained from the CHARGE consortium[71] and the analyses were performed using the ldsc tool (https://github.com/bulik/ldsc).

**Genetic risk scores**. We generated genetic risk scores (GRS) for Alzheimer's disease, amyotrophic lateral sclerosis (ALS), Parkinson's disease, bipolar disorder, schizophrenia, white matter lesions and tau-related phenotypes. The tau-related phenotypes, including tau and phosphorylated tau levels in cerebrospinal fluid, and progressive supranuclear palsy (PSP), were studied in relatively small sample and are therefore not appropriate for LD score regression. We extracted the lead genome-wide significantly associated SNPs and their effect estimates from the largest published GWA studies (Supplementary Data 10). For white matter lesions burden, effect estimate and standard errors were estimated from Z-statistics using the previously published formula[72]. The allele associated with an increased risk in corresponding traits was considered to be the effect allele. The weighted GRS was constructed as the sum of products of effect sizes as

weights and respective allele dosages from 1000 Genome imputed data of Rotterdam Study using R software version 3.2.5 (https://www.R-project.org). Variants with low imputation quality ($r^2 < 0.3$) were excluded. Subsequently, the GRS was tested for association with lateral ventricular volume in three cohorts of Rotterdam Study while adjusting for age, sex, total intracranial volume, age[2] and population stratification. The significance threshold for genetic risk score association was set to $p$-value $= 5 \times 10^{-3}$ (0.05/10) based on the number of genetic risk scores tested.

## Data availability

The summary statistics will be made available upon publication on the CHARGE dbGaP site under the accession number phs000930.v7.p1.

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

## Acknowledgements

See Supplementary Note 2 for information on funding sources.

## Author contributions

Conceived the study and drafted the manuscript: D.V., H.H.A., M.A.I., S.S., M.F. Performed statistical analyses: D.V., H.H.A., X.J., Q.Y., A.V.S., J.C.B., A.T., M.S., N.J.A., E.H., Y.S., M.L., M.B., S.T., J.Y., N.A.G., M.S.P., S.J.L., A.N., L.R.Y., and S.L. Acquired data: O.A.A., D.A., N.A., K.A., M.B., D.M.B., A.B., F.B., H.B., R.N.B., R.B., A.M.D., P.L.J., I.J.D., C.D., D.A.F., R.F.G., J.G., V.G., T.B.H., G.H., D.S.K., J.B.K., C.E.L., M.L., W.T.L., O.L.L., P.M., P.A.N., H.M., K.A.M., T.H.M., R.M., M.N., M.S.P., Z.P., B.M.P., K.R., C.L.S., R.S., P.S.S., P.J.S., S.S.S., D.J.S., A.T., A.G.U., M.C.V.H., M.W.V., W.W., T.W., A.V.W., K.W., M.J.W., H.T., W.K., D.A.B., J.W. J., T.P., J.A.W., H.S., P.S.S., A.V., H.J.G., J.I.R., C.M.D., L.J.L., S.S., M.A.I., M.F. All authors critically reviewed the manuscript for important intellectual content.

## Additional information

**Competing interests:** A.M.D. is a Founder of and holds equity in CorTechs Labs, Inc, and serves on its Scientific Advisory Board. He is a member of the Scientific Advisory Board of Human Longevity, Inc. and receives funding through research agreements with General Electric Healthcare and Medtronic, Inc. The terms of these arrangements have been reviewed and approved by UCSD in accordance with its conflict of interest policies. The remaining authors declare no competing interests.

Dina Vojinovic[1], Hieab H. Adams[1,2], Xueqiu Jian[3], Qiong Yang[4], Albert Vernon Smith[5,6], Joshua C. Bis[7], Alexander Teumer[8], Markus Scholz[9,10], Nicola J. Armstrong[11,12], Edith Hofer[13,14], Yasaman Saba[15], Michelle Luciano[16], Manon Bernard[17], Stella Trompet[18,19], Jingyun Yang[20,21], Nathan A. Gillespie[22], Sven J. van der Lee[1], Alexander Neumann[23], Shahzad Ahmad[1], Ole A. Andreassen[24], David Ames[25,26], Najaf Amin[1], Konstantinos Arfanakis[20,27,28], Mark E. Bastin[16,29,30], Diane M. Becker[31], Alexa S. Beiser[4], Frauke Beyer[32], Henry Brodaty[12,33], R. Nick Bryan[34], Robin Bülow[35], Anders M. Dale[36], Philip L. De Jager[37,38], Ian J. Deary[16], Charles DeCarli[39], Debra A. Fleischman[20,21,40], Rebecca F. Gottesman[41], Jeroen van der Grond[42], Vilmundur Gudnason[5,6], Tamara B. Harris[43], Georg Homuth[44], David S. Knopman[45], John B. Kwok[46,47], Cora E. Lewis[48], Shuo Li[4], Markus Loeffler[9,10], Oscar L. Lopez[49], Pauline Maillard[39], Hanan El Marroun[23,50], Karen A. Mather[12,51], Thomas H. Mosley[52], Ryan L. Muetzel[1,23], Matthias Nauck[53], Paul A. Nyquist[41], Matthew S. Panizzon[54], Zdenka Pausova[17,55], Bruce M. Psaty[7,56,57,58], Ken Rice[59], Jerome I. Rotter[60,61], Natalie Royle[16,29,30], Claudia L. Satizabal[62,63], Reinhold Schmidt[13], Peter R. Schofield[46,51], Pamela J. Schreiner[64], Stephen Sidney[65], David J. Stott[66], Anbupalam Thalamuthu[12], Andre G. Uitterlinden[67], Maria C. Valdés Hernández[16,29,30], Meike W. Vernooij[1,2], Wei Wen[12], Tonya White[2,23], A. Veronica Witte[32,68], Katharina Wittfeld[69], Margaret J. Wright[70], Lisa R. Yanek[31], Henning Tiemeier[23,71], William S. Kremen[54], David A. Bennett[20,21], J. Wouter Jukema[19,72], Tomas Paus[73,74], Joanna M. Wardlaw[16,29,30], Helena Schmidt[15], Perminder S. Sachdev[12,75], Arno Villringer[32,68], Hans Jörgen Grabe[69,76], W T Longstreth[56,77], Cornelia M. van Duijn[1,78], Lenore J. Launer[43], Sudha Seshadri[62,63], M Arfan Ikram[1,2,79] & Myriam Fornage[3]

[1]Department of Epidemiology, Erasmus MC University Medical Center, Rotterdam 3015 CN, The Netherlands. [2]Department of Radiology and Nuclear Medicine, Erasmus MC University Medical Center, Rotterdam 3015 CN, The Netherlands. [3]The University of Texas Health Science Center at Houston Institute of Molecular Medicine, Houston, TX 77030, USA. [4]Department of Biostatistics, School of Public Health, Boston University, Boston, MA 02118, USA. [5]Icelandic Heart Association, Kopavogur 201, Iceland. [6]Faculty of Medicine, University of Iceland, Reykjavik 101, Iceland. [7]Cardiovascular Health Research Unit, Department of Medicine, University of Washington, Seattle, WA 98101, USA. [8]Institute for Community Medicine, University Medicine Greifswald, Greifswald 17475, Germany. [9]Institute for Medical Informatics, Statistics and Epidemiology, University of Leipzig, Leipzig 04107, Germany. [10]LIFE Research Center for Civilization Diseases, University of Leipzig, Leipzig 04103, Germany. [11]Mathematics and Statistics, Murdoch University, Perth, WA 6150, Australia. [12]Centre for Healthy Brain Ageing, School of Psychiatry, UNSW, Sydney, NSW 2052, Australia. [13]Clinical Division of Neurogeriatrics, Department of Neurology, Medical University of Graz, Graz 8036, Austria. [14]Institute for Medical Informatics, Statistics and Documentation, Medical University of Graz, Graz 8036, Austria. [15]Gottfried Schatz research center, Institute for Molecular biology and biochemistry, Graz 8010, Austria. [16]Centre for Cognitive Ageing and Cognitive Epidemiology, Psychology, University of Edinburgh, Edinburgh EH8 9JZ, UK. [17]Research Institute of the Hospital for Sick Children, Toronto, Ontario M5G 1X8, Canada. [18]Section of Gerontology and Geriatrics, Department of Internal Medicine, Leiden University Medical Center, Leiden 2300 RC, The Netherlands. [19]Department of Cardiology, Leiden University Medical Center, Leiden 2300 RC, The Netherlands. [20]Rush Alzheimer's Disease Center, Rush University Medical Center, Chicago, IL 60612, USA. [21]Department of Neurological Sciences, Rush University Medical Center, Chicago, IL 60612, USA. [22]Virginia Institute for Psychiatric and Behavioral Genetics, Virginia Commonwealth University, Richmond, VA 23284, USA. [23]Department of Child and Adolescent Psychiatry/ Psychology, Erasmus University Medical Center Rotterdam, Rotterdam 3000 CB, The Netherlands. [24]Norwegian Centre for Mental Disorders Research (NORMENT), Institute of Clinical Medicine, University of Oslo, Oslo 0372, Norway. [25]National Ageing Research Institute, Melbourne, VIC 3052, Australia. [26]Academic Unit for Psychiatry of Old Age, University of Melbourne, Melbourne, VIC 3101, Australia.

[27]Department of Biomedical Engineering, Illinois Institute of Technology, Chicago, IL 60616, USA. [28]Department of Diagnostic Radiology and Nuclear Medicine, Rush University Medical Center, Chicago, IL 60612, USA. [29]Centre for Clinical Brain Sciences, University of Edinburgh & Brain Research Imaging Centre, University of Edinburgh, Edinburgh EH16 4SB, UK. [30]Division of Neuroimaging Sciences, Brain Research Imaging Centre, University of Edinburgh, Edinburgh EH16 4SB, UK. [31]Division of General Internal Medicine, Department of Medicine, Johns Hopkins School of Medicine, Baltimore, MD 21287, USA. [32]Department of Neurology, Max Planck Institute of Cognitive and Brain Sciences, Leipzig 04103, Germany. [33]Dementia Collaborative Research Centre – Assessment and Better Care, University of New South Wales, Sydney, NSW 2031, Australia. [34]The University of Texas at Austin Dell Medical School, Austin, TX 78705, USA. [35]Institute for Diagnostic Radiology and Neuroradiology, University Medicine Greifswald, Greifswald 17475, Germany. [36]Department of Cognitive Science, University of California, San Diego, La Jolla, CA 92093, USA. [37]Center for Translational and Computational Neuroimmunology, Department of Neurology, Columbia University Medical Center, New York, NY 10032, USA. [38]Cell Circuits Program, Broad Institute, Cambridge, MA 02142, USA. [39]Department of Neurology and Center for Neuroscience, University of California, Davis, CA 95817, USA. [40]Department of Behavioral Sciences, Rush University Medical Center, Chicago, IL 60612, USA. [41]Department of Neurology, Johns Hopkins School of Medicine, Baltimore, MD 21287, USA. [42]Department of Radiology, Leiden University Medical Center, Leiden 2300 RC, The Netherlands. [43]Laboratory of Epidemiology and Population Sciences, National Institute on Aging, Intramural Research Program, National Institutes of Health, Bethesda, MD 20892, USA. [44]Interfaculty Institute for Genetics and Functional Genomics, University Medicine Greifswald, Greifswald 17475, Germany. [45]Department of Neurology, Mayo Clinic, Rochester, MN 55905, USA. [46]School of Medical Sciences, University of New South Wales, Sydney, NSW 2052, Australia. [47]Brain and Mind Centre, The University of Sydney, Camperdown, NSW 2050, Australia. [48]The University of Alabama at Birmingham School of Medicine, Birmingham, AL 35294, USA. [49]Department of Neurology, University of Pittsburgh, Pittsburgh, PA 15213, USA. [50]Department of Psychology, Education and Child Studies, Erasmus University Rotterdam, Rotterdam, 3062 PA, The Netherlands. [51]Neuroscience Research Australia Randwick, Sydney, NSW 2031, Australia. [52]The University of Mississippi Medical Center, Jackson, MS 39216, USA. [53]Institute of Clinical Chemistry and Laboratory Medicine, University Medicine Greifswald, Greifswald 17475, Germany. [54]Department of Psychiatry, University of California, San Diego, La Jolla, CA 92093, USA. [55]Departments of Physiology and Nutritional Sciences, University of Toronto, Toronto, Ontario M5S 3E2, Canada. [56]Department of Epidemiology, University of Washington, Seattle, WA 98195, USA. [57]Department of Health Services, University of Washington, Seattle, WA 98195-7660, USA. [58]Kaiser Permanente Washington Health Research Institute, Seattle, WA 98101, USA. [59]Department of Biostatistics, University of Washington, Seattle, WA 98195, USA. [60]Institute for Translational Genomics and Population Sciences, Los Angeles Biomedical Research Institute at Harbor-UCLA Medical Center, Torrance, CA 90502, USA. [61]Department of Pediatrics, Harbor-UCLA Medical Center, Torrance, 90509 CA, USA. [62]Framingham Heart Study, Framingham, MA 01702, USA. [63]Department of Neurology, Boston University School of Medicine, Boston, MA 02118, USA. [64]University of Minnesota School of Public Health, Minneapolis, MN 55454, USA. [65]Kaiser Permanente Northern California Division of Research, Oakland, CA 94612, USA. [66]Institute of Cardiovascular and Medical Sciences, Faculty of Medicine, University of Glasgow, Glasgow G12 8QQ, UK. [67]Department of Internal Medicine, Erasmus MC University Medical Center, Rotterdam 3015, The Netherlands. [68]Faculty of Medicine, CRC 1052 Obesity Mechanisms, University of Leipzig, Leipzig 04103, Germany. [69]German Center for Neurodegenerative Diseases (DZNE), Rostock/Greifswald 17475, Germany. [70]Queensland Brain Institute, University of Queensland, Brisbane, QLD 4072, Australia. [71]Department of Psychiatry, Erasmus MC University Medical Center, Rotterdam 3015 GD, The Netherlands. [72]Einthoven Laboratory for Experimental Vascular Medicine, LUMC, Leiden 2300 RC, The Netherlands. [73]Rotman Research Institute, Baycrest, Toronto, Ontario M6A 2E1, Canada. [74]Departments of Psychology and Psychiatry, University of Toronto, Toronto, Ontario M5T 1R8, Canada. [75]Neuropsychiatric Institute, Prince of Wales Hospital, Randwick Sydney, NSW 2031, Australia. [76]Department of Psychiatry and Psychotherapy, University Medicine Greifswald, Greifswald 17475, Germany. [77]Department of Neurology, University of Washington, Seattle, WA 98195-6465, USA. [78]Nuffield Department of Population Health, University of Oxford, Oxford, OX37LF, UK. [79]Department of Neurology, Erasmus MC University Medical Center, Rotterdam 3015 GD, The Netherlands

