## [Peer Review File · Nature Communications]

Reviewer #1 (Remarks to the Author):

Vojinovic and colleagues present a GWAS study of human brain lateral ventricular (LV) volume. This is the first time that genetic loci have been identified for LV measures. The study is performed in a straightforward fashion in a two/three stage design that includes >23K aging individuals. GWAS has become mainstream and the excitement of discovering novel loci by itself is waning. At the same, recent developments of computational tools provide opportunity to examine the GWAS data in more detail and may provide more comprehensive biological insights. This study could use some of that novelty beyond what is currently presented.

Some specific points:

- I missed the description of SNP h^2 of the phenotype based on this study; how much of the variance is explained by these 7 loci?
- details are missing in the main text when discussing replication (page 8; stage 1-2). I assume that the direction of replication signal is the same but it is not mentioned.
- the sex specific analysis does not mention the sample size (males and females) in the text. Is the h^2 the same in males and females?
- the functional annotation and enrichment analysis is not the most advanced. It would be really helpful to partition the heritability by functional annotation; similarly, the use of recent tools (e.g. PrediXcan, SMR, TWAS) to impute the cis genetic component of gene expression would also highlight specific genes within known loci or may identify additional candidate genes.
- the authors present genetic correlations with multiple phenotypes and identify a negative correlation between LV and thalamus. It would really helpful to examine the correlation structure of available MRI phenotype data within this sample to confirm this genetic finding and/or identify other correlations at the phenotype level that do not yield a genetic correlation.
- the genetic risk score analysis between LV measures and other traits is interesting, specially the finding of an inverse correlation with CSF tau levels. However, it is likely that this effect is due to a few loci with major effect sizes instead of a true polygenic signal. Is the signal preserved when the known major tau loci are removed?
- Since this population is "older", the effect of apoe alleles could be suspected. However, there is no discussion about this in the paper.
- one of the challenges of large GWAS studies of complex traits is the heterogeneity of genetic signal. The overall sample consists of many different cohorts, including samples with non-europeans ancestry. How well did these African-American sample replicate the genetic findings? How well was the signal preserved between the EA cohorts?

Overall this is a valuable study that provides new insights. It is necessary that the summary stats of the stage 1,2 and 3 steps are made freely available to the scientific community.

Reviewer #2 (Remarks to the Author):

The authors describe an interesting study of the genetic contribution to the total volume of the lateral ventricles, through a GWAS design in a large meta-analysis of up to 26 healthy population cohorts. While the study is generally well performed and reports significant results that can provide valuable contributions that go beyond existing literature, there are a number of issues that reduce the general enthusiasm about this study.

Major points:

1. A measure of explained variance/heritability of the phenotypic trait is lacking. This is particularly important for lateral ventricle volume, as it has a relatively limited reproducibility in the generally used segmentation protocols.
2. The acquisition and segmentation methods are described in Supplementary Table 2 in a very patchy and non-systematic way. Also, segmentation appears to have been performed in a very heterogeneous manner across cohorts, and any analyses of comparability between methods is missing from the manuscript.
3. In general, the description of the imaging analysis that provided the phenotype for the current analysis in the Methods section lacks detail – there should be information about test-retest reliability of segmentation, correlation between methods, treatment of outliers, as well as plots of the distribution of the phenotype in the different cohorts in the supplement.
4. A sensitivity analysis using only the cohorts with a homogeneous method of phenotyping should be performed and presented.
5. The authors do not correct their phenotype for a measure of height/head size, even though ventricle size is known to be influenced by this (and there is nominally significant genetic correlation with height and ICV). This begs the question, how much of the results is actually due to body/head size. This is particularly relevant, as the authors find enrichment for cytoskeleton-related pathways, which also play important roles in height. An analysis with a height- or ICV-corrected phenotype should be added, and the variance explained by height/ICV should be determined.
6. In the Discussion section, more than 2 pages are used to describe the plausibility of the individual findings. This could easily be shortened by moving the info to a table. This would leave room for more important discussion points, like the differences between the findings for gene-based and functional analyses (e.g. GNA12 and GNA13), effects of height/ICV, explained variance, etc.
7. Discussion, page 13: you discuss the importance of the sex-specific analyses. However, this is not reflected by the findings or their description. This should be clarified.

8. Discussion, page 14: the discussion of the lack of overlap with neurological and psychiatric disorders is unclear. What is meant with '...suggesting that lateral ventricular volume is not disease-specific...'? Also, the sentence ending with '....genetic effects detectable by us due to lack of power.' contains two arguments that are intermingled. Importantly, a discussion about power aspects/heritability/measurement heterogeneity is lacking here.
9. More generally, a paragraph on limitations of the study is lacking in the Discussion section.
10. The study design is unclear – how was it determined, which cohorts to put into stage 1 and stage 2, respectively? This should be added, in the Methods section and Figure 1. Also, a justification of the design should be given in the last part of the Introduction section or the Discussion section – why go for discovery-replication and subsequent meta-analysis?

Additional points:

11. Introduction: the cited literature is generally quite dated, and – more importantly – is based on rather small sample sizes. Can the authors introduce more convincing evidence here?
12. Results, page 8: 'All stage 1 significant associations replicated' – please add the number of variants tested.
13. Results, page 8: 'The lead variants explained' – please indicate, how many variants were taken along in this estimation of variance explained?
14. Results, page 9: In the first paragraph, the authors describe analyses with childhood brain measures and some additional phenotypes, but the research question justifying these analyses is not provided. In addition, details about the 'PhenoScanner' analysis should be provided in the Methods section.
15. Results, page 9, first paragraph: What is the difference between 'top associations' and 'lead variants'? Please provide numbers of variants tested in these analyses in the text.
16. Results, page 9: The results of the sex-specific analysis are not adequately described: what was observed in this analysis, and how many variants were tested for heterogeneity?
17. Results, page 9, last line: why add information on TRIOBP here, but not on any of the other genes implicated?
18. Results, page 10, first line: what is meant by 'enrichment of GWA p-values in DNaseI hypersensitive sites'?
19. Results, page 10, line 2-3: can you specify 'The most significant association'?
20. Supplementary Figure 7 – this figure should be part of the main article instead of the supplement.
21. Results, page 10, line 2: should DNS be DHS?

22. Results, page 10, paragraph on Gene-based association analysis: please include information on the specific locus, in which the genes fall. This is important for the coherence of the text.
23. Results, page 10, paragraph on Gene-based association analysis: why describe functions of AP3M1 and GNA13, but not the other genes?
24. Throughout article: S1P is sometimes mentioned as SP1.
25. Table 2: Please add sample sizes.
26. Results, page 11, line 1-2: ‘...correlation was observed between genetic components of neurological and psychiatric diseases...’ – was this what was tested for?
27. Discussion, page 11: ‘...we found a significant correlation with genetic determinants of lateral ventricular and thalamus volumes.’ – sentence is unclear.
28. Discussion, page 11: There is no mention of the functional genomics analyses in the discussion of the 3q28 and 12q23.3 loci, while this should be an important source of information.
29. Discussion, page 12: How to explain the differences between the functional genomics and gene-based analysis results for e.g. 7q22.3? This should be discussed.
30. Discussion, page 13: ‘These data are consistent with our findings from the gene-enrichment analysis...’ – this is circular.
31. Methods, page 16, genome-wide association (GWA) analysis: Why did you transform the phenotype; what is meant by ‘(age2) when applicable’?
32. Methods, page 17: the description of the sex-specific analyses lacks detail.
33. Methods, page 18: was the analysis of the chromatin marks done in cis or genome-wide?
34. Methods, page 18, Functional annotation....: Describe the GARFIELD analysis in more detail, how were the T-values mentioned in the legend to supplementary figure 7 constructed?
35. Methods, page 18, Gene-based analyses: what was the reason for choosing the 10 kb area in the 5’ and 3’ UTRs? Please add references.
36. Methods, page 19, Pathway analysis: what was the p-value threshold of 5×10^{-5} based on?
37. Methods, page 19, Genetic correlation: ‘...obtained from the respective consortia....’ – which are those? Please add references/links.
38. Methods, page 20, line 1: What does RSI, RSII, and RSIII mean?
39. Methods, page 20: how did you adjust for population stratification?
40. Throughout the manuscript: please change ‘significant threshold’ into ‘significance threshold’.

Reviewer #3 (Remarks to the Author):

In this study, the authors present the first GWAS discovered common variants for lateral ventricular volumes using several studies that participated in the CHARGE consortium. The primary GWAS analyses appear conducted appropriately, following best practices and producing an interesting set of associations. The authors perform a number of secondary analysis to add context to their initial findings, but many are negative and at times presented contradictorily (described more below) which makes it hard to interpret the biological significance of these loci. Because of this and the lack of a true replication, the overall impact of the report may be lessened.

Some Specific Questions:

The authors use PhenoScanner, LDSC and GRS to show little to no overlap in their associations with neurological or psychiatric conditions. In the discussion, they lead off with: "Each novel significant association lies in a locus that has been previously related to brain disorders," and then discuss connections to brain disorders for each implicated locus. They then refer to the lack of connections to disease again (324-327) before concluding the paper with a statement about how this might help us understand disease (330-332). This seems contradictory. What analyses are being used to supports these disease connections described in the discussion? It appears as if they are anecdotes from the published literature, but anecdotes that seem counter to the results presented throughout.

Continuing with this point, the authors seem to want to discuss or make implications for these results in terms of neurological or psychiatric disorders, but given the population samples which appear (?) relatively free from disease and the lack of genetic connections to disease, would this paper not be more naturally framed in terms of the genetics of healthy (or perhaps just) neurological aging? Do the connections to CSF tau levels fit with this? Perhaps this could be discussed as a parallel interpretation.

The study is presented as a two stage-meta analysis with a discovery and a replication, but all samples were a part of the CHARGE consortium. Presumably the entirety of this data was in existence when this study was conducted. How did the authors divide the studies into a discovery and replication set? Why should the "final" meta-analysis not be considered the discovery data set? That seems like a more straightforward description, although it does expose the lack of a true replication.

Why did the authors limit the GRS analysis to only the top SNPs for each disorder they considered? As such, it seems redundant with the PhenoScanner cataloged variant associations, but weaker because it does not take advantage of the full data set (only the RS studies).

I am not sure I agree that the Gene-based analyses provide independent discoveries. The associations are less strong ($1e-6$) for these tests than for the lead SNP alone ($1e-7$ to $1e-16$). This makes me think these results are simply repackaging the single SNP test, paying a cost for a few more degrees of freedom, and then testing with a less stringent p-value threshold. Is there any evidence for these tests that collapsing across the gene is bringing independent statistical signal to light?

Minor Comments:

The authors aim to model away age effects in the regressions. To what extent do the effect of the top variants vary systematically with age (as is observed with the h^2 estimates)? There appears to be trending heterogeneity (Supplementary Table 5) for the second cohort which also seems to include cohorts that span a larger age range. Are these trends indicative of age dependent effects of the variants?

What was the effect of the known Alzheimer's APOE variants on this phenotype? Would that provide clues into the potential impact of un-annotated or early stage disease as a mediator of these results?

In Supplementary Table 3, could the authors provide the model fit by each study as well? They list only the software, but describing the model and covariates would be helpful.

In table 2 I find it compelling that, although not strictly significant, the trends for the genetic correlations with other volumetric measures are essentially consistent with what one might expect, that is genes increasing ventricular volume correlate with decreasing nuclei volumes and/or increasing total ICV (also infant head circumference).

180-181: "The lead variants explained 1.3% of variance in lateral ventricular volume in the Rotterdam Study III"

How was this percentage calculated? Is it meant to represent an out of sample prediction? Was this data included in the meta-analysis?

183-185: "Most of the lead variants showed consistent direction of effect with stages 3..."

Could these results be put into a supplementary table? How many had concordant directions? Was it a significant amount (Fisher's Test, Binomial proportion, etc)?

189-190: "The results of sex-stratified GWA analysis..."

Could you include the total sample sizes here?

206-207: "Functional enrichment analysis revealed ubiquitous enrichment of GWA p-values in DNaseI hypersensitivity sites (DNS) (Supplementary Fig. 7)"

This figure is hard to read, could the results also be put into a supplementary table?

Textual Suggestions:

162-163: "Briefly, we applied a three-stage design"

256:258: "Interestingly, the lead SNP of the 12q23.3 (rs12146713), mapping intron of NUA1, is predicted to be in one of the most deleterious variants in the human genome (top 1%, Supplementary Table 7, CADD=21.5).

This sentence is hard to understand. Does the lead SNP map onto the intron? Does the CADD score refer to the lead SNP?

305-311: “When examining the genetic overlap between lateral ventricular volume and various traits using currently available GWA summary data, we found no genetic correlation with neurological and psychiatric diseases such as Alzheimer’s disease, Parkinson’s disease, cerebral small vessel disease, bipolar disorder, and schizophrenia, suggesting either that lateral ventricular volume is not disease-specific and implying that these conditions are influenced by different gene sets, or that any sharing is not due to genetic effects detectable by us due to lack of power”

This sentence is a bit of a mouthful. I don’t follow the conclusion about the negative finding is beyond the lack of power

Response to Reviewers: NCOMMS-17-27386 (Vojinovic et al.)

We would like to express our thanks to the reviewers for their constructive comments and criticisms. Our specific responses to reviewer concerns are described below (reviewer comments are in bold; our responses are in normal text).

Reviewer #1 (Remarks to the Author):

Vojinovic and colleagues present a GWAS study of human brain lateral ventricular (LV) volume. This is the first time that genetic loci have been identified for LV measures. The study is performed in a straightforward fashion in a two/three stage design that includes >23K aging individuals. GWAS has become mainstream and the excitement of discovering novel loci by itself is waning. At the same, recent developments of computational tools provide opportunity to examine the GWAS data in more detail and may provide more comprehensive biological insights. This study could use some of that novelty beyond what is currently presented.

Some specific points:

- I missed the description of SNP h² of the phenotype based on this study; how much of the variance is explained by these 7 loci?

We apologize for this omission. We now provide SNP-based h² of the lateral ventricular volume and percentage of variance explained by 7 lead variants in the revised version of the manuscript. SNP-based heritability was estimated at 0.20 (SE=0.02) by LD score regression suggesting that common genetic variants represent a substantial fraction of overall genetic component of variance. The lead variants explained 1.5 % of total variance of lateral ventricular volume. We have discussed this in relevant sections of the Results (page 10), Discussion (page 15) and Method (page 20).

- details are missing in the main text when discussing replication (page 8; stage 1-2). I assume that the direction of replication signal is the same but it is not mentioned.

We have now provided missing information in the relevant section of the revised version of the manuscript. All variants replicated in stage 2 meta-analysis with the same direction of effect. Please see page 8.

- the sex-specific analysis does not mention the sample size (males and females) in the text. Is the h² the same in males and females?

We have added the sample size for sex-stratified analysis in the revised version of the manuscript on page 9. As suggested by reviewer we calculated the SNP-based h² for males and females and observed higher SNP-based heritability estimates in women (0.19, SE=0.04) than in men (0.15, SE=0.05). We have made changes accordingly in the results section of the manuscript on pages 9, 10 and 15.

- the functional annotation and enrichment analysis is not the most advanced. It would be really helpful to partition the heritability by functional annotation; similarly, the use of recent tools (e.g. PrediXcan, SMR, TWAS) to impute the cis genetic component of

gene expression would also highlight specific genes within known loci or may identify additional candidate genes.

Complying with the reviewer's suggestion, we have partitioned heritability based on functional annotation as described in the revised version of the manuscript on pages 10, 15 and 20. The analysis revealed significant threefold enrichment of SNPs within 500 bp of highly active enhancers where 17% of SNPs accounted for 54% of the heritability. Furthermore, significant enrichment was also found for various histone marks.

Additionally, we integrated the functional data generated by the GTEx project and our meta-analysis results by using MetaXcan which is an extension of PrediXcan method modified to use summary statistic data from a meta-analysis. The MetaXcan analysis identified altered expression of 2 genes in the brain to be implicated in lateral ventricular volume at Bonferroni-corrected significance. In addition to *TRIOBP* which was also discovered in the single variant analysis, a novel candidate gene *MRPS16* was implicated. Mutations in *MRPS16* gene have previously been related to agenesis/hypoplasia of corpus callosum and enlarged ventricles. We have discussed this in relevant sections of the revised version of the manuscript on pages 11, 14 and 21.

- the authors present genetic correlations with multiple phenotypes and identify a negative correlation between LV and thalamus. It would really helpful to examine the correlation structure of available MRI phenotype data within this sample to confirm this genetic finding and/or identify other correlations at the phenotype level that do not yield a genetic correlation.

We have explored the phenotypic correlation between the lateral ventricular volume and available MRI phenotype data within the sample as suggested by the reviewer. The results are illustrated in the Table below. We confirmed our genetic finding as the negative correlation between the lateral ventricular volume and thalamus was also observed at the phenotype level. Even though not strictly significant, we also observed the trends for the genetic correlation with other volumetric measures including the negative genetic correlation with nuclei volume. These results are also confirmed at the phenotype level. Furthermore, we observed the positive correlation between lateral ventricular volume and other cavities of the ventricular system. We have discussed this in a relevant section of the revised version of the manuscript on page 12.

Table. Phenotypic correlation between lateral ventricular volume and MRI phenotypes. Partial correlation coefficient is calculated while controlling for the total intracranial volume.

	Thalamus	Caudate	Putamen	Pallidum	3rd.Ventricle	4th.Ventricle	Brain Stem	Hippocampus	Amygdala	CSF	Accumbens.area	5th.Ventricle
CHS	-0.34	0.21	-0.18	-0.22	0.67	0.36	-0.33	-0.48	-0.27	0.54	-0.41	0.00
RS1	-0.24	0.28	-0.17	-0.31	0.66	0.26	-0.30	-0.36	-0.12	0.57	-0.24	0.19
RS2	-0.37	0.31	-0.17	-0.35	0.69	0.31	-0.28	-0.40	-0.21	0.57	-0.25	0.12
RS3	-0.36	0.18	-0.23	-0.28	0.65	0.24	-0.27	-0.35	-0.24	0.57	-0.32	0.18

- the genetic risk score analysis between LV measures and other traits is interesting, specially the finding of an inverse correlation with CSF tau levels. However, it is likely that this effect is due to a few loci with major effect sizes instead of a true polygenic signal. Is the signal preserved when the known major tau loci are removed?

We would like to point to the reviewer that we have used only 3 known SNPs genome-wide significantly associated with CSF tau levels to construct the risk score (Supplementary Table 14). We further explored whether the effect is the combined effect of 3 SNPs or if it is coming from one of the three. As illustrated in the Table below, the association is driven by one SNP. We have mentioned this on page 22 in the ‘genetic risk score’ section of the methods. We have also discussed this in a relevant section of the results on page 12.

Table. Tau SNPs used for genetic risk score analysis and their association with lateral ventricular volume

SNP	Estimate	SE	P-value
rs514716	0.008	0.014	5.76E-01
rs9877502	0.045	0.010	4.76E-06
rs769449	0.006	0.014	6.42E-01

- Since this population is "older", the effect of apoe alleles could be suspected. However, there is no discussion about this in the paper.

We thank the reviewer for pointing this out. We have checked the association of *APOE* alleles with lateral ventricular volume in our study. We have found no association between *APOE* ϵ 4 (Z-score=-0.171, p-value=0.86) or *APOE* ϵ 2 (Z-score=-0.238, p-value=0.81) alleles and lateral ventricular volume. This could be explained by the fact that we focused on general population which is relatively free from relevant disease as participants with stroke and dementia at the time of MRI were excluded from the analyses. We have discussed this in the relevant section of the Discussion on page 15.

- one of the challenges of large GWAS studies of complex traits is the heterogeneity of genetic signal. The overall sample consists of many different cohorts, including samples with non-europeans ancestry. How well did these African-American sample replicate the genetic findings? How well was the signal preserved between the EA cohorts?

Indeed our overall sample consists of both European (EA) and African-American (AA) ancestry cohorts. Even though we have included different ancestries, due to a very small sample size of AA sample (N=1,488), all significant associations were mainly driven by EA. With regard to reviewer’s question about the concordance of the signal across the EA cohorts, for each of 7 lead genome-wide significant variants, we now provide a scatterplot of Z-scores by cohort sample size in Supplementary Material. Please see Supplementary Fig. 11. The direction of effect size across the EA cohorts was generally concordant and showed no evidence of single cohort driving the associations. We have clarified this in the relevant section of Results on page 9.

- Overall this is a valuable study that provides new insights. It is necessary that the summary stats of the stage 1,2 and 3 steps are made freely available to the scientific community.

We agree with the reviewer. The summary statistics will be made freely available upon the publication on the CHARGE dbGaP site that was put together for that purpose.

Reviewer #2 (Remarks to the Author):

The authors describe an interesting study of the genetic contribution to the total volume of the lateral ventricles, through a GWAS design in a large meta-analysis of up to 26 healthy population cohorts. While the study is generally well performed and reports significant results that can provide valuable contributions that go beyond existing literature, there are a number of issues that reduce the general enthusiasm about this study.

Major points:

1. A measure of explained variance/heritability of the phenotypic trait is lacking. This is particularly important for lateral ventricle volume, as it has a relatively limited reproducibility in the generally used segmentation protocols.

We apologize for this omission. We now provide SNP-based heritability estimate of the lateral ventricular volume in the revised version of the manuscript. SNP-based heritability was estimated at 0.20 (SE=0.02) by LD score regression suggesting that common variants represent a substantial fraction of overall genetic component of variance. We have discussed this in the relevant sections of the Results (page 10), Discussion (page 15) and Method (page 20).

2. The acquisition and segmentation methods are described in Supplementary Table 2 in a very patchy and non-systematic way. Also, segmentation appears to have been performed in a very heterogeneous manner across cohorts, and any analyses of comparability between methods is missing from the manuscript.

Complying with the reviewer's suggestion, we now provide a description of the acquisition and segmentation methods in a more systematic way in the revised version of the supplementary material. Please see Supplementary Table 2.

Regarding the reviewer's question about the comparability between the methods, the comparability between the visual and volumetric scales was evaluated carefully and reported previously.¹ It has been shown that agreement between the visual and volumetric rating for lateral ventricular volume is high ($r^2=0.7$), suggesting that both methods were estimating lateral ventricular size robustly and accurately. Regarding the automated segmentation method, we have performed a reproducibility study in a subset of participants for Rotterdam Study. The assessment of consistency of lateral ventricular volume across the different versions of software (freesurfer v4.5, v5.1, and v6.0) and across time, revealed high intraclass correlation (ICC>0.98). We have discussed this in the relevant section of the revised version of the manuscript. Please see pages 17 and 18.

Furthermore, assuming the heterogeneity of effect sizes across the studies, association results of participating cohorts were combined using a fixed-effect sample-size weighted Z-score meta-analysis. This approach controls the differences in the phenotype across the cohorts and weights the Z statistic from each study by its sample size. Our purpose was to identify new associations rather than accurately estimating the effect size of well-validated variants, which would need to account for possible between-population heterogeneity. This is a standard approach in imaging genetics as demonstrated by previously published studies.^{2,3}

¹ Carmichael, Owen T., et al. "Ventricular volume and dementia progression in the Cardiovascular Health Study." *Neurobiology of aging* 28.3 (2007): 389-397.

² Hibar, Derrek P., et al. "Novel genetic loci associated with hippocampal volume." *Nature communications* 8 (2017): 13624.

3. In general, the description of the imaging analysis that provided the phenotype for the current analysis in the Methods section lacks detail – there should be information about test-retest reliability of segmentation, correlation between methods, treatment of outliers, as well as plots of the distribution of the phenotype in the different cohorts in the supplement.

Complying with the reviewer's request, we now provide more details for the imaging analysis in the relevant section of Methods in the revised version of the manuscript on pages 17 and 18. Each cohort used a validated visual grading or validated automated segmentation method to quantify lateral ventricular volume (please see the answer to question 2). Furthermore, as requested by the reviewer, we also performed a reproducibility study (please see the answer to question 2). Additionally, we provided the distribution plots of untransformed and transformed phenotype in the different cohorts. Please see Supplementary Material, Supplementary Fig. 17.

4. A sensitivity analysis using only the cohorts with a homogeneous method of phenotyping should be performed and presented.

As per reviewer's request, we performed a sensitivity analysis by combining the summary statistics data of cohorts with a homogeneous method of phenotyping (N=11,287). Of the 7 originally significant hits, two variants show genome-wide association (see Table below), while the other 5 show suggestive association ($p\text{-value} < 10^{-6}$). The effect estimates were concordant in direction with the effects obtained in the original meta-analysis. These results suggest that despite heterogeneity association signals also come from cohorts that are not phenotyped in the same way, suggesting the robustness of our findings.

Table. Lead 7 SNPs and their association results in a subset of the sample with a homogeneous method of phenotyping.

MarkerName	Allele1	Allele2	Freq1	N	Zscore	P-value
12:106476805	t	c	0.91	11287	-5.70	1.17E-08
3:190620146	a	g	0.42	11287	-5.56	2.67E-08
10:21878144	a	t	0.31	11287	-5.02	5.19E-07
11:111077725	a	g	0.67	11287	4.93	8.30E-07
22:38110450	t	c	0.59	11287	-4.83	1.34E-06
7:2760334:C_CT	d	i	0.72	10343	-4.82	1.41E-06
16:87225101	a	g	0.59	11287	4.47	7.70E-06

Furthermore, we have generated PM-plots for each of the 7 lead variants in order to analyze heterogeneity across the studies included in the original meta-analysis. The plots display the posterior probability that the effect exists and the p-values in each study. As illustrated in Supplementary Fig. 12, both studies with visual and volumetric method of phenotype assessment have the effect, confirming again robustness of our findings. We have discussed this in the revised version of the manuscript on page 9. The PM-plot for the lead variant on 3q28 is illustrated below.

³ Verhaaren, Benjamin FJ, et al. "Multiethnic Genome-Wide Association Study of Cerebral White Matter Hyperintensities on MRICLINICAL PERSPECTIVE." *Circulation: Genomic and Precision Medicine* 8.2 (2015): 398-409.

Figure. PM plots for the lead variant at 3q28.

The x-axis displays the m-values (the posterior probability that the effect exists) and the y-axis represents $-\log_{10}(p)$ -values) in each study. The red dots denote that the study has an effect ($m \geq 0.9$), and green dots represent studies whose effect is uncertain ($0.1 < m < 0.9$). The studies with the visual method of phenotyping include ARIC_V3_AA, ARIC_V3_EA, ASPS, CHS_EA and CHS_AA, while in other cohorts volumetric methods of phenotyping were used. The results suggest that despite heterogeneity signals also come from cohorts that are not phenotyped in the same way, confirming again the robustness of our findings.

5. The authors do not correct their phenotype for a measure of height/head size, even though ventricle size is known to be influenced by this (and there is nominally significant genetic correlation with height and ICV). This begs the question, how much of the results is actually due to body/head size. This is particularly relevant, as the authors find enrichment for cytoskeleton-related pathways, which also play important roles in height. An analysis with a height- or ICV-corrected phenotype should be added, and the variance explained by height/ICV should be determined.

We apologize for not being clear. We agree with the reviewer that ventricle size is known to be influenced by the head size and that there is a nominally significant correlation between the lateral ventricular volume and intracranial volume. Therefore, all performed association analyses were adjusted for intracranial volume. We have clarified this in the method section of the revised version of the manuscript (page 18).

We have also determined the percentage of variance in lateral ventricular volume explained by ICV in 4493 participants from 3 population-based cohorts to be ranged between 11.5% and 16%.

6. In the Discussion section, more than 2 pages are used to describe the plausibility of the individual findings. This could easily be shortened by moving the info to a table. This would leave room for more important discussion points, like the differences between the findings for gene-based and functional analyses (e.g. GNA12 and GNA13), effects of height/ICV, explained variance, etc.

Complying with the reviewer's suggestion, we shortened the discussion section used to describe the plausibility of the individual findings and added the text regarding the SNP-based heritability, percentage of variance explained and functional analyses. The changes have been made on pages 13-15.

7. Discussion, page 13: you discuss the importance of the sex-specific analyses. However, this is not reflected by the findings or their description. This should be clarified.

As per the reviewer's request, we have clarified the results of sex-stratified analysis in the relevant section of discussion in the revised version of the manuscript. Please see page 14. We have clarified that we did not observe sex-specific differences for the 7 lead variants as both males and females were contributing to the association signal. Additionally, we have mentioned that we have observed only one suggestive association in men.

8. Discussion, page 14: the discussion of the lack of overlap with neurological and psychiatric disorders is unclear. What is meant with '...suggesting that lateral ventricular volume is not disease-specific...'? Also, the sentence ending with '.....genetic effects detectable by us due to lack of power.' contains two arguments that are intermingled. Importantly, a discussion about power aspects/heritability/measurement heterogeneity is lacking here.

We apologize for not being clear. We have rephrased the paragraph on page 15. We have also added the text regarding the SNP-based heritability, the percentage of variance explained, functional analyses and measurement heterogeneity in the relevant sections of the revised version of the manuscript. Please see pages 14-16. With regard to reviewer's question about power aspects, our stage 1 analysis was sufficiently powered to detect variants with both medium and small effect size for $\alpha=0.0000001$.

9. More generally, a paragraph on limitations of the study is lacking in the Discussion section.

We have added the following text in the relevant section of Discussion in the revised version of the manuscript on page 16.

“The strengths of our study are the large sample, population-based design and the use of quantitative MRI. Our study also has several limitations. Despite the effort to harmonize phenotype assessment, the methods used to quantify lateral ventricular volume differ across cohorts. Because of this phenotypic heterogeneity, association results of participating cohorts were combined using a sample-size weighted meta-analysis, thus limiting discussion on effect sizes. Secondly, phenotypic heterogeneity may have caused the loss of statistical power. However, despite heterogeneity in the phenotype assessment, the association signals were coming from several studies irrespective of the phenotype assessment, which suggests robustness of our findings. Another limitation of our study was that our sample comprised predominantly of EA participants, therefore, our findings may not be generalizable to other ethnic groups.”

10. The study design is unclear – how was it determined, which cohorts to put into stage 1 and stage 2, respectively? This should be added, in the Methods section and Figure 1. Also, a justification of the design should be given in the last part of the Introduction section or the Discussion section – why go for discovery-replication and subsequent meta-analysis?

We apologize for not being clear. We have clarified the study design in the Method section of the revised version of the manuscript (page 17) and in Supplementary Figure 1 where we

provide the overview of study design. The stage 1 comprised studies that contributed summary statistic data before a certain deadline which was set before inspecting the data and was not influenced by the results of the GWA meta-analysis.

With regard to reviewer's question about the study design, we have applied commonly used two-stage approach,^{4,5,6} followed by joint analysis strategy that combines information across the stages and provides greater power. We have clarified this in the relevant section of the Introduction in the revised version of the manuscript. Please see page 8.

Additional points:

11. Introduction: the cited literature is generally quite dated, and – more importantly – is based on rather small sample sizes. Can the authors introduce more convincing evidence here?

We have tried to be very exhaustive and inclusive with references to genetic papers, however, the number of genetic studies is very limited. Upon suggestion of the reviewer, more convincing references on the association of lateral ventricular volume with brain-related disorders are provided. Please see page 7.

12. Results, page 8: 'All stage 1 significant associations replicated' – please add the number of variants tested.

We have added the number of variants tested in the replication phase in the result section of the revised version of the manuscript on page 8.

13. Results, page 8: 'The lead variants explained' – please indicate, how many variants were taken along in this estimation of variance explained?

We have indicated that 7 lead variants that were taken along in the estimation of variance in the relevant section of Results. Please see page 10.

14. Results, page 9: In the first paragraph, the authors describe analyses with childhood brain measures and some additional phenotypes, but the research question justifying these analyses is not provided. In addition, details about the 'PhenoScanner' analysis should be provided in the Methods section.

We have clarified that the analyses in the children's cohort were performed in order to explore whether the lead variants found to be associated with lateral ventricular volume in adulthood also have an effect in early life, childhood. Please see the relevant section of results in the revised version of the manuscript on page 9. Additionally, we explored whether any of the lead variants or their proxies had a pleiotropic association with other traits or diseases by using the PhenoScanner database. The details about the PhenoScanner database designed to the cross-reference genetic variants with a broad range of phenotypes were provided in the Method section of the revised version of the manuscript on page 20.

15. Results, page 9, first paragraph: What is the difference between 'top associations'

⁴ Lambert, Jean-Charles, et al. "Meta-analysis of 74,046 individuals identifies 11 new susceptibility loci for Alzheimer's disease." *Nature genetics* 45.12 (2013): 1452.

⁵ Kinnersley, Ben, et al. "Genome-wide association study identifies multiple susceptibility loci for glioma." *Nature communications* 6 (2015): 8559.

⁶ Nalls, Mike A., et al. "Large-scale meta-analysis of genome-wide association data identifies six new risk loci for Parkinson's disease." *Nature genetics* 46.9 (2014): 989.

and ‘lead variants’? Please provide numbers of variants tested in these analyses in the text.

We apologize for the confusion caused by using different terms while referring to the same variants. We have clarified that by top associations we meant lead variants. Please see page 9.

16. Results, page 9: The results of the sex-specific analysis are not adequately described: what was observed in this analysis, and how many variants were tested for heterogeneity?

We now describe the results of the sex-stratified analysis in the revised version of the manuscript on page 9. The analyses were performed on a sample of 10,358 men and 12,872 women. None of the 15,660,719 variants with heterogeneous test statistics between men and women surpassed genome-wide significance level. However, the top suggestive association was observed between an indel at 4q35.2 and lateral ventricular volume in men but not in women.

17. Results, page 9, last line: why add information on *TRIOBP* here, but not on any of the other genes implicated?

Indeed we added information on *TRIOBP* but not on any of the other genes implicated as *TRIOBP* is the only gene which is differently expressed in brain tissue. We have clarified this in the results section of the revised version of the manuscript on page 10. Interestingly, the same gene was also identified in the MetaXcan analysis associating the expression of *TRIOBP* in brain with lateral ventricular volume (p -value= 3.2×10^{-6}). The results of this analysis are described in the revised version of the manuscript on page 11.

18. Results, page 10, first line: what is meant by ‘enrichment of GWA p-values in DNaseI hypersensitive sites’?

We apologize for not being clear. What we meant is that SNPs associated with lateral ventricular volume at a p -value $< 10^{-5}$ were more often located in genomic regions that were marked by histone modifications (carrying H3K36me3 and H3K9ac marks) and DNaseI hypersensitivity sites (hotspots). We have clarified this in the relevant section of the Results in the revised version of the manuscript on page 11.

19. Results, page 10, line 2-3: can you specify ‘The most significant association’?

We have clarified how the significance threshold was calculated based on the number of annotations and set at 4.97×10^{-5} as described in the Method section on page 21.

20. Supplementary Figure 7 – this figure should be part of the main article instead of the supplement.

Complying with the reviewer’s suggestion we now include Supplementary Figure 7 as a part of the main article as Figure 3 (page 35).

21. Results, page 10, line 2: should DNS be DHS?

We thank the reviewer for pointing this out. We have replaced DNS with DHS at page 11.

22. Results, page 10, paragraph on Gene-based association analysis: please include information on the specific locus, in which the genes fall. This is important for the coherence of the text.

We have added the information on the specific locus in which the genes fall in the revised version of Supplementary Material. Please see Supplementary Table 10. However, as pointed out by the reviewer #3, the associations for genes are less strong than for individual variants. We agree with this point that the results cannot be interpreted as independent discoveries and have rephrased the gene-based analysis results in the revised version of the manuscript on page 11.

23. Results, page 10, paragraph on Gene-based association analysis: why describe functions of AP3M1 and GNA13, but not the other genes?

As already discussed in the response to the reviewer's question 22, we have rephrased the paragraph about the gene-based results. Please see page 11.

24. Throughout article: S1P is sometimes mentioned as SP1.

We thank the reviewer for pointing this out. We have made corrections in the revised version of the manuscript on pages 6 and 16.

25. Table 2: Please add sample sizes.

We have added sample size to Table 2 on page 37 of the revised version of the manuscript.

26. Results, page 11, line 1-2: '...correlation was observed between genetic components of neurological and psychiatric diseases...' – was this what was tested for?

We thank the reviewer for pointing this out. We have rewritten this sentence in the revised version of the manuscript. Please see page 12.

27. Discussion, page 11: '...we found a significant correlation with genetic determinants of lateral ventricular and thalamus volumes.' – sentence is unclear.

We apologize for not being clear. We have rephrased the sentence. Please see page 13 of the revised version of the manuscript.

28. Discussion, page 11: There is no mention of the functional genomics analyses in the discussion of the 3q28 and 12q23.3 loci, while this should be an important source of information.

We have provided the additional information in the relevant section of Discussion on page 13. The lead variant at 12q23.3 locus is among the top 1% of the most deleterious variants in the human genome (CADD=21.5) and was located in enhancer region (Supplementary Table 7). With regard to 3q28 locus, most variants have no functional consequences (Supplementary Table 7).

29. Discussion, page 12: How to explain the differences between the functional genomics and gene-based analysis results for e.g. 7p22.3? This should be discussed.

Functional genomics focused on the dynamic aspects such as gene transcription. The analysis revealed that lead variant at 7p22.3 was in active chromatin state and associated with differential expression of *GNAI2*. On the other hand, the gene-based analysis was based on all variants irrespective of their functional consequence. The analysis also revealed the association of the *GNAI2* gene and lateral ventricular volume. We are not sure which differences the reviewer is referring to.

30. Discussion, page 13: ‘These data are consistent with our findings from the gene-enrichment analysis....’ – this is circular.

We have rephrased the sentence in the relevant section of Discussion in the revised version of the manuscript on page 14.

31. Methods, page 16, genome-wide association (GWA) analysis: Why did you transform the phenotype; what is meant by ‘(age²) when applicable’?

Lateral ventricular volume has a positively skewed distribution and the transformation was performed to obtain approximately normal distribution. We mention this now in the methods section (page 18). We have also clarified that age² when applicable refers to including age squared as a variable if it showed significant evidence of association with lateral ventricular volume. Please see the page 18.

32. Methods, page 17: the description of the sex-specific analyses lacks detail.

As suggested by the reviewer we have provided additional details regarding the sex-stratified analysis in the revised version of the manuscript on page 19.

33. Methods, page 18: was the analysis of the chromatin marks done in cis or genome-wide?

We have used the RoadMap data in order to explore the significant combinatorial interactions between different chromatin marks in their special context. Roadmap dataset was generated genome-wide but we did a lookup 100bp around the lead SNPs. Details of the RoadMap are provided in the paper by Ernest et al.⁷

34. Methods, page 18, Functional annotation....: Describe the GARFIELD analysis in more detail, how were the T-values mentioned in the legend to supplementary figure 7 constructed?

We have described the GARFIELD analysis in more detail in the relevant section of Methods on page 21. We have also clarified that T-values mentioned in the legend to Supplementary Figure 7 now included as Figure 3 refer to GWA *p*-value thresholds and have also provided which *p*-value thresholds were used.

⁷ Ernst, Jason, and Manolis Kellis. "ChromHMM: automating chromatin-state discovery and characterization." *Nature methods* 9.3 (2012): 215.

35. Methods, page 18, Gene-based analyses: what was the reason for choosing the 10 kb area in the 5' and 3' UTRs? Please add references.

We have included variants within 10 kb of the 5' and 3'UTR in order to maintain regulatory variants. We have clarified this and have added the reference now in the revised version of the manuscript on page 21.

36. Methods, page 19, Pathway analysis: what was the p-value threshold of 5×10^{-5} based on?

We have clarified what the significance threshold was for pathway-based analysis in the relevant section of methods on page 22. The significance threshold was based on number of multiple correlated pathways. We have also provided the reference.

37. Methods, page 19, Genetic correlation: ‘...obtained from the respective consortia....’ – which are those? Please add references/links.

We have added the name of consortium and have incorporated the reference on page 22 of the revised version of the manuscript.

38. Methods, page 20, line 1: What does RSI, RSII, and RSIII mean?

We have clarified that RSI, RSII, and RSIII refer to three cohorts of the Rotterdam study. Please see changes in the relevant section of methods on page 23.

39. Methods, page 20: how did you adjust for population stratification?

Population stratification was controlled for by including principal components derived from genome-wide genotype data. We have clarified this in the relevant section of methods in the revised version of the manuscript on page 18.

40. Throughout the manuscript: please change ‘significant threshold’ into ‘significance threshold’.

We have made changes according to the reviewer’s suggestion throughout the manuscript.

Reviewer #3 (Remarks to the Author):

In this study, the authors present the first GWAS discovered common variants for lateral ventricular volumes using several studies that participated in the CHARGE consortium. The primary GWAS analyses appear conducted appropriately, following best practices and producing an interesting set of associations. The authors perform a number of secondary analysis to add context to their initial findings, but many are negative and at times presented contradictorily (described more below) which makes it hard to interpret the biological significance of these loci. Because of this and the lack of a true replication, the overall impact of the report may be lessened.

Some Specific Questions:

The authors use PhenoScanner, LDSC and GRS to show little to no overlap in their

associations with neurological or psychiatric conditions. In the discussion, they lead off with: “Each novel significant association lies in a locus that has been previously related to brain disorders,” and then discuss connections to brain disorders for each implicated locus. They then refer to the lack of connections to disease again (324-327) before concluding the paper with a statement about how this might help us understand disease (330-332). This seems contradictory. What analyses are being used to supports these disease connections described in the discussion? It appears as if they are anecdotes from the published literature, but anecdotes that seem counter to the results presented throughout.

We understand the confusion of the reviewer. Indeed the results look contradictory, however, we would like to point out to the reviewer that LDSC is based on the entire genome and GRS is also based on multiple variants, while the results of PhenoScanner are per individual genetic variants. The association of lead variants and their proxies according to the PhenoScanner database are displayed in the Table below. So while the top SNPs may be connected with brain disorders, it may not be true genome-wide. The disease connections with genome-wide significant genetic loci we have retrieved from the published literature, so these are established associations and we are not performing any analyses to support these disease connections. We have provided the references to the original publications.

Table. Association of lead SNPs and their proxies ($r^2 > 0.7$, $p\text{-value} < 5E-08$) with various trait and diseases according to PhenoScanner database

SNP	Proxy rsID	r^2	Trait	PMID	Beta	SE	P
rs35587371	rs11012732	0.88	Meningioma	21804547	-0.378	0.049	2E-14
rs35587371	rs12770228	0.81	Meningioma	21804547	NA	NA	4.72E-11
rs34113929	rs9877502	0.77	Alzheimers disease biomarkers	23562540	0.052	0.009	5E-09
rs34113929	rs9877502	0.77	Cerebrospinal fluid CSF tau	23562540	NA	NA	4.98E-09

Continuing with this point, the authors seem to want to discuss or make implications for these results in terms of neurological or psychiatric disorders, but given the population samples which appear (?) relatively free from disease and the lack of genetic connections to disease, would this paper not be more naturally framed in terms of the genetics of healthy (or perhaps just) neurological aging? Do the connections to CSF tau levels fit with this? Perhaps this could be discussed as a parallel interpretation.

We thank the reviewer for pointing this out. Indeed we wanted to make implication for the results in terms of neurological and psychiatric disorders, however, we agree with the reviewer that the paper would be more naturally framed in terms of the genetics of neurological aging. We have made changes in the relevant part of the discussion in the revised version of the manuscript. Please see page 16.

The study is presented as a two stage-meta analysis with a discovery and a replication, but all samples were a part of the CHARGE consortium. Presumably the entirety of this data was in existence when this study was conducted. How did the authors divide the studies into a discovery and replication set? Why should the “final” meta-analysis not be considered the discovery data set? That seems like a more straightforward description, although it does expose the lack of a true replication.

We apologize for not being clear about the study design. The discovery stage comprised studies that contributed summary statistic data before a certain deadline which was set before inspecting the data and was not influenced by the results of the GWA meta-analysis. We clarified this in the relevant section of revised version of the manuscript on page 17.

With regard to reviewer's question about the study design, we have applied a commonly used two-stage approach,^{8,9,10} followed by joint analysis strategy that combines information across the stages and provides greater power.¹¹ We have clarified this in the relevant section of the Introduction in the revised version of the manuscript. Please see page 8. Even though additional loci reached the genome-wide significance only when including both discovery and replication samples, they should still be considered as highly probable findings but not a replicated loci and would still require independent replication.

Why did the authors limit the GRS analysis to only the top SNPs for each disorder they considered? As such, it seems redundant with the PhenoScanner cataloged variant associations, but weaker because it does not take advantage of the full data set (only the RS studies).

We can extend the GRS to include genome-wide SNPs, however, in the absence of any evidence for pleiotropic effects using LD score regression (which is a better algorithm to identify genetic correlations), we wonder how useful it would be to develop a polygenic risk score. We already observe dilution of effects when top variants are bundled in a GRS together (Supplementary Table 15 and 16). If the reviewer insists, we will extend the GRS.

I am not sure I agree that the Gene-based analyses provide independent discoveries. The associations are less strong (1e-6) for these tests than for the lead SNP alone (1e-7 to 1e-16). This makes me think these results are simply repackaging the single SNP test, paying a cost for a few more degrees of freedom, and then testing with a less stringent p-value threshold. Is there any evidence for these tests that collapsing across the gene is bringing independent statistical signal to light?

We completely agree with the reviewer's point therefore in the revised version of the manuscript we use the gene-based results only for pathway analysis in VEGAS tool. Please see pages 11 and 21. We do not mention these as independent discoveries.

Minor Comments:

The authors aim to model away age effects in the regressions. To what extent do the effect of the top variants vary systematically with age (as is observed with the h2 estimates)? There appears to be trending heterogeneity (Supplementary Table 5) for the second cohort which also seems to include cohorts that span a larger age range. Are these trends indicative of age dependent effects of the variants?

⁸ Lambert, Jean-Charles, et al. "Meta-analysis of 74,046 individuals identifies 11 new susceptibility loci for Alzheimer's disease." *Nature genetics* 45.12 (2013): 1452.

⁹ Kinnersley, Ben, et al. "Genome-wide association study identifies multiple susceptibility loci for glioma." *Nature communications* 6 (2015): 8559.

¹⁰ Nalls, Mike A., et al. "Large-scale meta-analysis of genome-wide association data identifies six new risk loci for Parkinson's disease." *Nature genetics* 46.9 (2014): 989.

¹¹ Skol, Andrew D., et al. "Joint analysis is more efficient than replication-based analysis for two-stage genome-wide association studies." *Nature genetics* 38.2 (2006): 209.

Indeed, as observed by the reviewer, the cohorts included in the stage 2 span a larger age range. As suggested by reviewer we have explored if the lead variants showed age-dependent effects. We have compared the effect estimates of lead variants and mean age of cohorts with a homogeneous method of phenotyping (a subset of total sample). The results of the analyses are illustrated in the revised version of Supplementary Material (Supplementary Fig. 4-10). We observed that nearly all effects were stable across the age range 50 to 85 years, but one of the 7 significant loci showed an effect related to the mean age of cohort as shown in the Figure below. The effect size for variant mapped to 10p12.31 locus was correlated with mean age of the cohort ($r=0.498$, $p=0.025$). The effect was near to zero at younger mean ages and larger at older mean ages. This has been discussed in the relevant sections of Results and Discussion in the revised version of the manuscript on pages 9 and 14.

Figure. Plot of effect size and mean age of cohorts with homogeneous phenotyping method for lead variant at 10p12.31 locus. Each number point denotes a cohort (1- CARDIA-AA, 2- SYS, 3-SHIP-trend, 4-CARDIA-EA, 5-SHIP, 6-VETSA, 7-RSIII, 8-FHS, 9-LIFE-Adult, 10-RSII, 11-OATS, 12-LBC1936, 13-ARIC-V5-AA, 14-PROSPER, 15-AGES, 16-ARIC-V5-EA, 17-MAS, 18-RSI, 19- RUSH-ROSMAP Batch 2, 20-RUSH-ROSMAP Batch 1). The plot includes SD of age range for each cohort.

In Supplementary Table 3, could the authors provide the model fit by each study as well? They list only the software, but describing the model and covariates would be a helpful.

Complying with the reviewer's suggestion, we have provided the covariates for each study in the revised version of the Supplementary Table 3. Each study performed analysis using the following model: $Y = \beta_0 + \beta_G \text{SNP} + \beta_C C$. Where Y is the lateral ventricular volume, β_0 is the intercept, SNP is the dosage of genetic variant coded additively and C is the vector of covariates including age, age^2 (if significant), sex, total intracranial volume, study-specific confounders, PCs, familial relationship. We have clarified this in Supplementary Table 3.

In table 2 I find it compelling that, although not strictly significant, the trends for the genetic correlations with other volumetric measures are essentially consistent with what

one might expect, that is genes increasing ventricular volume correlate with decreasing nuclei volumes and/or increasing total ICV (also infant head circumference).

We thank the reviewer for pointing this out. We have discussed this in the relevant section of Results and Discussion of the revised version of the manuscript on pages 12 and 15.

180-181: “The lead variants explained 1.3% of variance in lateral ventricular volume in the Rotterdam Study III”. How was this percentage calculated? Is it meant to represent an out of sample prediction? Was this data included in the meta-analysis?

The percentage of variance explained by the lead variants was calculated based on the genetic data of RSIII participants. Indeed, the RSIII was part of overall meta-analysis. Additionally, we have also calculated the percentage of variance explained by using Pearson’s phi coefficient squared as explained in Draisma et al.¹² The total proportion of variance in lateral ventricular volume explained by 7 lead variants was estimated to be 1.5%. We have clarified this in the revised version of the manuscript on page 20.

183-185: “Most of the lead variants showed consistent direction of effect with stages 3...” Could these results be put into a supplementary table? How many had concordant directions? Was it a significant amount (Fisher’s Test, Binomial proportion, etc)?

As suggested by the reviewer, we have incorporated the association results from the children’s cohort into the Supplementary Table 5. The percentage of lead variants showing consistent direction of effect with stage 3 was 85.7 % (6 out of 7, binomial p-value=0.05). We have made changes in the relevant section of results in the revised version of the manuscript (page 9).

189-190: “The results of sex-stratified GWA analysis...Could you include the total sample sizes here?”

We have included the total sample size for sex-stratified analysis in the relevant section of the revised version of the manuscript (page 9).

**206-207: “Functional enrichment analysis revealed ubiquitous enrichment of GWA p-values in DNaseI hypersensitivity sites (DNS) (Supplementary Fig. 7)”
This figure is hard to read, could the results also be put into a supplementary table?**

Complying with the reviewer’s suggestion, we now provide the results of functional enrichment analysis in a supplementary material. Please see Supplementary Table 9 of the revised version of the Supplementary material.

Textual Suggestions:

162-163: “Briefly, we applied a three-stage design”

We have removed the sentence. Please see page 8.

256:258: “Interestingly, the lead SNP of the 12q23.3 (rs12146713), mapping intron of

¹² Draisma, Harmen HM, et al. "Genome-wide association study identifies novel genetic variants contributing to variation in blood metabolite levels." *Nature communications* 6 (2015): 7208.

NUAK1, is predicted to be in one of the most deleterious variants in the human genome (top 1%, Supplementary Table 7, CADD=21.5).

This sentence is hard to understand. Does the lead SNP map onto the intron? Does the CADD score refer to the lead SNP?

We have rewritten the sentence. Please see page 13.

305-311: “When examining the genetic overlap between lateral ventricular volume and various traits using currently available GWA summary data, we found no genetic correlation with neurological and psychiatric diseases such as Alzheimer’s disease, Parkinson’s disease, cerebral small vessel disease, bipolar disorder, and schizophrenia, suggesting either that lateral ventricular volume is not disease-specific and implying that these conditions are influenced by different gene sets, or that any sharing is not due to genetic effects detectable by us due to lack of power”

This sentence is a bit of a mouthful. I don’t follow the conclusion about the negative finding is beyond the lack of power.

We have rewritten the sentence. Please see page 15.

Reviewer #1 (Remarks to the Author):

The authors responded well to the questions raised by the reviewers. The paper has certainly improved.

Yet, I would have liked a more robust discussion and reflection about the findings. For example, no further information is given about the actual contribution of non-European ancestry samples except to say that most of the signal is coming from European ancestry samples. There is no further discussion beyond the mentioning of the "single intergenic SNP" that shows "suggestive" evidence for involvement in CSF tau levels and LV volumes. (BTW if the finding is suggestive at best, why report it?)

Reviewer #2 (Remarks to the Author):

The authors did a very good job in responding to the comments from all three reviewers. I'm satisfied with their handling of my comments in the rebuttal and the manuscript and have no further points to raise.

Reviewer #3 (Remarks to the Author):

I have Included my comments as an attachment

Response to Reviewers: NCOMMS-17-27386 (Vojinovic et al.)

We would like to express our thanks to the reviewers for their constructive comments and criticisms. Our specific responses to reviewer concerns are described below (reviewer comments are in bold; our responses are in normal text).

Reviewer #1 (Remarks to the Author):

The authors responded well to the questions raised by the reviewers. The paper has certainly improved.

Yet, I would have liked a more robust discussion and reflection about the findings. For example, no further information is given about the actual contribution of non-European ancestry samples except to say that most of the signal is coming from European ancestry samples. There is no further discussion beyond the mentioning of the "single intergenic SNP" that shows "suggestive" evidence for involvement in CSF tau levels and LV volumes. (BTW if the finding is suggestive at best, why report it?)

We thank the reviewer for the suggestions which improved the manuscript. With regard to reviewer's suggestion to provide further information about the contribution of non-European ancestry sample, we now discuss this in the relevant section of Discussion on page 17. Although we made an effort to include cohorts of both European and African American ancestry, our study comprised predominately of individuals of European origin (22,045 individuals of European ancestry and 1,488 individuals of African American ancestry). Given the disparity in sample size, it is difficult to distinguish whether any inconsistency in results between the 2 groups stems from true genetic differences or from differential power to detect genetic effects. Indeed, this is exemplified by the plots of Z-scores (Supplementary Figure 11) showing that direction of effect in African American cohorts is often inconsistent with the direction of effect in European cohorts. However, the same inconsistency can be observed with European cohorts of equally small sample size. This inconsistency may be due to small sample size rather than ethnic background but we cannot rule out that racial-ethnic specific effects may exist. This limitation underscores the need for expanding research studies in non-European populations. With regard to the suggestive association of single SNP involved in CSF tau levels and LV volumes, we agree with the reviewer and we have removed it from the discussion.

Reviewer #2 (Remarks to the Author):

The authors did a very good job in responding to the comments from all three reviewers. I'm satisfied with their handling of my comments in the rebuttal and the manuscript and have no further points to raise.

We thank the reviewer for the in-depth-comments, suggestions, and corrections, which improved the manuscript.

Reviewer #3 (Remarks to the Author):

The authors have done a commendable job addressing many of my concerns and I appreciate the extra clarity, explanation and analysis provided throughout, however, I still have an important concern regarding the interpretation of some of the results and

(lack of) broader framing in the discussion. Also, some additional points of clarity and context are suggested for the text added throughout during revision.

My major concerns linger around my first critique and the authors' response:

1. The authors use PhenoScanner, LDSC and GRS to show little to no overlap in their associations with neurological or psychiatric conditions. In the discussion, they lead off with: "Each novel significant association lies in a locus that has been previously related to brain disorders," and then discuss connections to brain disorders for each implicated locus. They then refer to the lack of connections to disease again (324-327) before concluding the paper with a statement about how this might help us understand disease (330-332). This seems contradictory. What analyses are being used to supports these disease connections described in the discussion? It appears as if they are anecdotes from the published literature, but anecdotes that seem counter to the results presented throughout.

We understand the confusion of the reviewer. Indeed the results look contradictory, however, we would like to point out to the reviewer that LDSC is based on the entire genome and GRS is also based on multiple variants, while the results of PhenoScanner are per individual genetic variants. The association of lead variants and their proxies according to the PhenoScanner database are displayed in the Table below. So while the top SNPs may be connected with brain disorders, it may not be true genome-wide.

I have a few concerns with this response and its lack of redress in the discussion. I agree with the authors that the three analyses they present represent an investigation on three different scales: LDSC / polygenic, GRS / oligogenic, GWAS hits / monogenic, and that is fine. These are not entirely independent as the monogenic is nested within the oligogenic is nested within the polygenic, but they do summarize the data from different perspectives. The strongest associations they observed are in the most specific scale (phenoScanner results described in the table below) and with a specific few phenotypes. Some more guidance in how to interpret (potential) discrepancies across scales is necessary.

To that point, I think it would greatly benefit the paper to see a paragraph or two in the discussion dedicated to a more nuanced assessment of these different sources of information about genetic overlap between LV and "brain disorders" and their support for/against or limitations in addressing what I see as one of the authors two key motivations presented for pursuing this research:

"Elucidating the genetic contribution to inter-individual variation in lateral ventricular volume can thus provide important insights and better understanding of the complex genetic architecture of brain structures and related neurological and psychiatric disorders."

Is the evidence strongly supportive, inconclusive, strongly negative, varying, etc. and what is needed to remedy?

For example, a recent report found little evidence for genetic overlap, on multiple scales, between hippocampal volume measurements and schizophrenia (10.1038/nn.4228), where similar motivational arguments have been made. Are the patterns here similar or

different to what was observed with that study? What might that say about this motivation?

Given the only overlap was found at the level of a few single SNPs should I interpret a genetic overlap between “brain disorders” and LV as the rule or the exception?

To that point, the authors suggest there is a connection to “brain disorders” as if it is general or the group of phenotypes considered is homogeneous. I disagree with both. The PhenoScanner showed connections to meningeal tumors and some biochemical markers for a neurological disease (AD). There were no highlighted connections to psychiatric disorders or other neurological disorders. Am I to believe that the connections reported support a genetic overlap between LV and all “brain disorders”? This would seem to imply that the genetics of meningeal tumors or neurological disorder biomarkers are informative for psychiatric conditions, is this a reasonable inference? Given some specificity of genetic effects with respect to neurological and psychiatry disease (10.1038/ng.3406, www.biorxiv.org/content/early/2017/09/06/048991), I am not sure interpreting results in terms of “brain disorders” in general is supported by the current presentation and description of the results and think some nuance with respect to the heterogeneity of “brain disorders” is appropriate. This could be discussed.

What are the limitations that might prevent the different scales of data from speaking more strongly to the overlap in different domains? For example, the LDSC point estimates for some correlations have a “non-significant” p-value, but the trends/point estimates themselves are not always close to 0. How should I think about these results in light of the power of the study? Are they definitive one way or the other? Similarly, the phenoScanner database only seems to include genome-wide significant hits. This could limit the picture of overlap at the single SNP level, is there a way to address this in future studies? Might this limit the breadth of associations currently observed?

One important strength of this study, that I see, is that although it may or may not be able to speak definitively to all of these questions currently, by providing the first large, well-powered, appropriately conducted GWAS of LV that is a more or less a typically aging population free of disease confounding, it will provide a resource (summary stats) that is critically needed to fuel more studies and more definitively address some of these lingering questions through an expanded consensus (just my opinion). In this light, I see this paper as a first step in resolving these issues and the inability to provide a definitive statement is expected, and so some guidance for future directions, given any limitations, could be important here. Would the authors agree with this? Perhaps this is a strength in light of some earlier inconsistencies.

The disease connections with genome-wide significant genetic loci we have retrieved from the published literature, so these are established associations and we are not performing any analyses to support these disease connections. We have provided the references to the original publications.

I agree they are retrieved from the published literature, but do not agree that that makes them “established associations” and I do not see the lack of analysis as a strength, but rather a weakness (we don’t get a sense of how likely these connections are to occur by chance), in their presentation. As an example:

Line 301: “whereas the locus at 22q13.1 has been linked to schizophrenia. 38, 39”

The authors tested the hypothesis of a connection between locus 22 and schizophrenia through phenoScanner and did not observe a connection, but provide this contradictory anecdote as an “established association.” What goal does this serve? What should I conclude about the analysis within this paper?

Citation 39, the “established” supporting genotype-phenotype association, is a single linkage analysis from 1994, in 39 families with a reported LOD score of 1.54. Both the title of that paper (which calls the study “part 1”) and abstract (“This finding is of sufficient interest to warrant further investigation through collaborative studies”) suggest the original authors do not consider this linkage definitive. Why does this paper reflect the current scientific consensus about that locus’ involvement in schizophrenia? I could understand highlighting an inconsistency with an “established association,” but would take that to mean an inconsistency with “a broader scientific consensus” rather than “an isolated suggestive study” and it would need to be discussed in the context of the strengths and limitations of the analyses presented within this report that focus on exactly this question. So, this particular anecdote appears to make a connection between LV genetic effects and schizophrenia but it does so by side stepping the research presented within this paper. I find that misleading and think it serves as an unnecessary tangent that detracts from or obfuscates the actual data analysis presented within. I would encourage the authors to carefully reconsider the goals of presenting genotype-phenotype anecdotes such as this in the discussion with respect to their reflection of broader consensus and recast their discussion of them in light of the consistency/inconsistency with and strengths/weakness of the design of their own analytic framework, or remove them.

Table. Association of lead SNPs and their proxies ($r^2 > 0.7$, $p\text{-value} < 5E-08$) with various trait and diseases according to PhenoScanner database

SNP	Proxy rsID	r2	Trait	PMID	Beta	SE	P
rs35587371	rs11012732	0.88	Meningioma	21804547	-0.378	0.049	2E-14
rs35587371	rs12770228	0.81	Meningioma	21804547	NA	NA	4.72E-11
			Alzheimers disease				
rs34113929	rs9877502	0.77	biomarkers	23562540	0.052	0.009	5E-09
			Cerebrospinal fluid CSF				
rs34113929	rs9877502	0.77	tau	23562540	NA	NA	4.98E-09

In general, I would like to see a bit more relationship to their main motivations, rather than lists of anecdotal connections. I think it is a missed opportunity.

We now provide the following paragraph in the relevant section of the discussion dedicated to genetic overlap between lateral ventricular volume and neurological or psychiatric disorders (page 16).

“However, while studying genetic overlap of lateral ventricular volume and various neurological or psychiatric disorders at multiple levels (LD score regression/polygenic, GRS/oligogenic, GWA hits/monogenic), we found evidence that some single genetic variants have pleiotropic effect on lateral ventricular volume and biochemical markers for a neurological disease (AD) or meningioma (Supplementary Table 6), while no evidence was found for genetic overlap with other neurological or psychiatric disorders (Table 2, Supplementary Table 15). The pattern of association between lateral ventricular volume and psychiatric disorder i.e. schizophrenia on multiple scales is similar to the findings of Franke et al. who evaluated the association of various subcortical brain volumes and schizophrenia and

reported no evidence of genetic overlap.⁴⁹ Even though our study does not provide a definite statement regarding the relationship between lateral ventricular volume and neurological or psychiatric disorders, it lays the foundation for future studies which should disentangle whether lateral ventricular volume is genetically related or unrelated to various neurological and psychiatric disorders (e.g. result from reverse causation). Novel insights may be revealed by improving the power of the studies, studying homogeneous samples with harmonized phenotype assessment methods along with evaluation of common and rare variants.”

We agree with the reviewer that interpreting results in terms of brain disorders is not supported by the current results. Therefore, we have omitted this term and have provided more specific interpretation of the results. With regard to reviewer’s comment about the locus 22q13.1, we agree with the reviewer’s point and we have removed this from the relevant section of the discussion on page 14.

2. Why did the authors limit the GRS analysis to only the top SNPs for each disorder they considered? As such, it seems redundant with the PhenoScanner cataloged variant associations, but weaker because it does not take advantage of the full data set (only the RS studies).

We can extend the GRS to include genome-wide SNPs, however, in the absence of any evidence for pleiotropic effects using LD score regression (which is a better algorithm to identify genetic correlations), we wonder how useful it would be to develop a polygenic risk score. We already observe dilution of effects when top variants are bundled in a GRS together (Supplementary Table 15 and 16). If the reviewer insists, we will extend the GRS.

I do not insist, but see above regarding a discussion on the implications of different levels of analysis.

We have incorporated reviewer’s suggestion regarding the discussion on the implications of different levels of analysis. Please see reply to question 1.

Comments on additional text/analysis:

3. Line 184-187: The effect size for the lead variant mapped to 10p12.31 locus was correlated with mean age of the cohort ($r = 0.50$, p -value = 0.03) (Supplementary Fig. 4). No correlation was found for the other lead variants (Supplementary Fig. 5-10)

The potential age x genotype interaction is interesting. Thank you for this additional analysis.

We thank the reviewer for the suggestion to check for potential age x genotype interaction.

4. Line 189-192: The direction of effect size across the EA cohorts for the 7 lead variants was generally concordant and showed no evidence of any single cohort driving the associations (Supplementary Fig. 11).

I would recommend plotting these in terms of betas and standard errors as using the z-score scales effects by sample size which will introduce hard-to-interpret variability. Perhaps consider a forest plot. They could be scaled to standard deviation units if the measurement scales vary across studies. Some color coordination by major methodological difference could be helpful.

We agree with the reviewer that plotting betas and standard errors would be more informative. However, cohorts included in our study used different scales to assess lateral ventricular volume including visual scale ranging from 0 to 3 or 0 to 9 or volume in mm³. Thus, plotting betas would not help interpreting our finding as betas are not comparable. Because of this, we were prompted to perform sample-size weighted meta-analysis. We have discussed this as a limitation of the current study in the relevant section of discussion on page 17. As suggested by the reviewer the measurement could have been scaled to standard deviation units as the measurement scales vary across the studies. However, this had to be done in each cohort prior to the analysis.

With regard to reviewer's suggestion that some color coordination by major methodical differences could be helpful, we now provide revised versions of plots. Please see Supplementary Figure 11.

5. Line 184-187: *Despite the different methods of phenotyping across the cohorts, the cohorts with different phenotyping methods contributed to the association signals (Supplementary Fig. 12).*

“contributed to the association signal” is kind of an odd phrase. All of the studies contribute because they are all in the meta-analysis. Are the authors suggesting that despite methodological variabilities in acquisition, the genetic effects are qualitatively similar? Or there is a limited heterogeneity in effects?

We have rewritten the sentence in the relevant section of the revised version of the manuscript on page 9. These results are suggesting that despite methodological variabilities in the acquisition of the data, there is limited heterogeneity in effects.

6. Line 220-221: *In brain tissue, the alternate allele of this SNP was associated with higher expression of TRIOBP (Supplementary Fig. 15).*

..suggesting higher expression was association with larger/smaller ventricles?

Higher expression of *TRIOBP* was associated with smaller lateral ventricles. We have now provided missing information in the relevant section of the revised version of the manuscript on page 10.

7. Line 228-229: *Significant enrichment was also found for histone marks*
Line 235-236: *more often located in genomic regions harboring histone marks*

Could some more context be added to the “histone marks.” This is a diverse group of annotations that can be related to regulatory elements, active gene bodies, repressed chromatin, and in different tissues.

We now provide following information to the histone marks in the relevant sections of results on pages 10 and 11.

“Significant enrichment was also found for histone marks including H3K27ac (which indicate enhancer and promoter regions), H3K9ac (which highlights promoters), H3K4me3 (which indicates promoters/transcription starts), and H3K4me1 (which highlights enhancers) (Supplementary Table 8).” (page 10)

“Functional enrichment analysis using regulatory regions from the ENCODE and Roadmap projects using the GWAS Analysis of Regulatory or Functional Information Enrichment with LD correction (GARFIELD) method revealed that SNPs associated with lateral ventricular volume at p -value threshold $<10^{-5}$ were more often located in genomic regions harboring histone marks (H3K9ac (associated with promoters) and H3K36me3 (associated with transcribed regions))...” (page 11)

With regard to reviewer’s question about the tissues, for partitioned heritability analysis we used “full baseline model” as described in Finucane *et al*¹. for 24 main annotations that are not specific to any cell type, whereas for histone marks and other annotations that differed among cell types, different cell types were combined in one annotation. Thus, this limited our ability to discuss the tissues.

8. Line 234-236: that SNPs associated with lateral ventricular volume at p -value threshold $<10^{-5}$ were more often located in genomic regions harboring histone marks and DNaseI hypersensitivity sites (DHS) more often than what?

Chance? Repressed chromatin? A permuted background? A little guiding text here would be helpful.

We have clarified that the SNPs associated with lateral ventricular volume at p -value threshold $<10^{-5}$ were more often located in genomic regions harboring histone marks and DNaseI hypersensitivity sites than a permuted background. Please see relevant section of results on page 11 of the revised version of the manuscript.

9. Line 257-262: This finding was also confirmed at the phenotype level (Supplementary Table 13). Weaker genetic overlap was observed with infant head circumference ($\rho_{genetic} = 0.28$, p -value = 8.7×10^{-3}), intracranial volume ($\rho_{genetic} = 0.35$, p -value = 9×10^{-3}), height ($\rho_{genetic} = -0.14$, p -value = 5.7×10^{-3}),

In the methods section the authors state the analyses adjusted for ICV (Lines 415-416) but in supplementary table 3 (where the models are described per cohort) it appears ~20% of the sample did not account for ICV in their regression analysis. This is also inconsistent with the response to reviewer 2, point 5. These genetic correlations seem to imply the SNP effects in this GWAS are not independent of body size, which could complicate the interpretation. Why were all studies not adjusted for ICV in a consistent manner? Do these genetic correlations persist when considering only the studies that did adjust for ICV? Should this be viewed as a limitation?

Indeed, as reported in the Supplementary Table 3 not all studies applied adjustment for intracranial volume while running linear regression analysis. However, studies that did not adjust for intracranial volume in the analysis used visual grading scale for assessment of phenotype and intracranial volume is accounted for in lateral ventricular volume assessment. Therefore, although they did not control for intracranial volume in regression analysis, ICV has been taken into account in phenotype assessment. We have clarified this in the Supplementary Table 3.

¹ Finucane, Hilary K., et al. "Partitioning heritability by functional annotation using genome-wide association summary statistics." *Nature genetics* 47.11 (2015): 1228.

10. Line 329-330: However, the SNP-based heritability estimates were slightly higher in females. This may be explained by the differences in sample size in male and female-specific analyses.

Is the difference significant or of a sufficient magnitude to be meaningful given the size of error in these estimates? Is the implication that SNP heritability increases monotonically with sample size or there is lower precision in the sex-specific analyses?

Indeed, the difference between the estimates is not of significant magnitude given the size of the error in these estimates. As SNP-based heritability was calculated using unbiased LD score regression method, SNP heritability won't increase monotonically with sample size, there could just be lower precision in the sex-specific analysis. We have clarified this in the relevant section of Discussion on page 15.

11. The study is presented as a two stage-meta analysis with a discovery and a replication, but all samples were a part of the CHARGE consortium. Presumably the entirety of this data was in existence when this study was conducted. How did the authors divide the studies into a discovery and replication set? Why should the “final” meta-analysis not be considered the discovery data set? That seems like a more straightforward description, although it does expose the lack of a true replication.

We apologize for not being clear about the study design. The discovery stage comprised studies that contributed summary statistic data before a certain deadline which was set before inspecting the data and was not influenced by the results of the GWA meta-analysis. We clarified this in the relevant section of revised version of the manuscript on page 17. With regard to reviewer's question about the study design, we have applied a commonly used two-stage approach,^{8,9,10} followed by joint analysis strategy that combines information across the stages and provides greater power.¹¹ We have clarified this in the relevant section of the Introduction in the revised version of the manuscript. Please see page 8. Even though additional loci reached the genome-wide significance only when including both discovery and replication samples, they should still be considered as highly probable findings but not a replicated loci and would still require independent replication.

I may have a different interpretation of the Skol paper (11) than the authors. As I see it the authors (Skol et al) are showing that the joint meta-analysis is more powerful than a two stage – in essence, it contains all of and more information than a design treating the two studies separately. One could say then that the two-stage presentation adds nothing over the presentation of joint analysis – it is extraneous. This is a minor point and I only meant it as a way to simplify. I understand the field is used to this kind of two stage plus meta-analysis presentation and do not see it as a major issue. I leave it to authors as to whether they want to keep or not. As the authors say some loci only reach significance in the meta-analysis and so some statement about the lack of/need for replication in a limitation in the discussion is sufficient. This was more the main point.

We have added the statement about the need for replication for loci that reached the genome-wide significance in the combined meta-analysis to the relevant section of the Discussion on page 17.

Reviewer #3 (Remarks to the Author):

Thanks to the authors for a nice response and set of clarifications and revisions. I don't have any further comments.

Response to Reviewers: NCOMMS-17-27386 (Vojinovic et al.)

Reviewer #3 (Remarks to the Author):

Thanks to the authors for a nice response and set of clarifications and revisions. I don't have any further comments.

We thank the reviewer for the in-depth-comments, suggestions, and corrections, which improved the manuscript.